# Towards Scalable Topological Regularizers

**Hiu-Tung Wong**[1,†]**, Darrick Lee**[2,†]**, Hong Yan**[1,3]
[1]Centre for Intelligent Multidimensional Data Analysis, Science and Technology Park, Hong Kong
[2]School of Mathematics, University of Edinburgh, UK
[3]Department of Electrical Engineering, City University of Hong Kong, Kowloon, Hong Kong
hiutung@innocimda.com, darrick.lee@ed.ac.uk, h.yan@cityu.edu.hk

## Abstract

Latent space matching, which consists of matching distributions of features in latent space, is a crucial component for tasks such as adversarial attacks and defenses, domain adaptation, and generative modelling. Metrics for probability measures, such as Wasserstein and maximum mean discrepancy, are commonly used to quantify the differences between such distributions. However, these are often costly to compute, or do not appropriately take the geometric and topological features of the distributions into consideration. Persistent homology is a tool from topological data analysis which quantifies the multi-scale topological structure of point clouds, and has recently been used as a topological regularizer in learning tasks. However, computation costs preclude larger scale computations, and discontinuities in the gradient lead to unstable training behavior such as in adversarial tasks. We propose the use of *principal persistence measures*, based on computing the persistent homology of a large number of small subsamples, as a topological regularizer. We provide a parallelized GPU implementation of this regularizer, and prove that gradients are continuous for smooth densities. Furthermore, we demonstrate the efficacy of this regularizer on shape matching, image generation, and semi-supervised learning tasks, opening the door towards a scalable regularizer for topological features.

## 1 Introduction

Latent space matching is a fundamental task in deep learning. Quantifying differences in latent representations and optimizing the network accordingly enables applications such as adversarial attack and defenses (Yu et al., 2021; Madaan et al., 2020; Lin et al., 2020), domain adaptation (Sun et al., 2016; Sun & Saenko, 2016; Long et al., 2017; Xu et al., 2019) and few-shot learning (Schonfeld et al., 2019; Xu et al., 2022; Mondal et al., 2023). Unsupervised training frameworks such as Generative Adversarial Networks (GANs) (Goodfellow et al., 2014) are fundamentally built on this concept, which is the primary framework considered throughout this article.

**Topological Features of Latent Representations.** The *manifold hypothesis* states that real-world high dimensional data sets are often concentrated about lower dimensional submanifolds. Recent work has empirically verified that image datasets such as CIFAR-10 and ImageNet satisfy a *union of manifolds hypothesis* (Brown et al., 2022), where the intrinsic dimension of connected components may be different. In the context of GANs, correctly learning the geometric and topological properties of these lower-dimensional structures enable meaningful interpolation in data space by traversing network latent spaces. Such properties are crucial to generalization ability (Zhou et al., 2020; Wang et al., 2021b) and generation quality (Zhu et al., 2023; Katsumata et al., 2024). Furthermore, topological metrics based on *persistent homology* provide highly effective evaluation metrics for GANs (Zhou et al., 2021; Khrulkov & Oseledets, 2018; Barannikov et al., 2021; Charlier et al., 2019).

Standard approaches to latent space matching use metrics on probability measures such as the Wasserstein distance and maximum mean discrepancy (MMD) metrics, which *implicitly* take topological features into consideration. In particular, the measures (and thus all topological properties) become

---

[†]Equal contribution.

equivalent when the distance is trivial. However, GANs are not guaranteed to reach a global minimum, and often converge to local saddle points (Berard et al., 2019; Liang & Stokes, 2019). When measures are a finite distance apart, their topological properties may be distinct. Motivated by the above work, which demonstrates that topological similarity between real and generated distributions is a critical component of GAN performance, we propose the use of a topological regularizer which *explicitly* measures the difference between topological features at non-equilibrium states.

**Persistent Homology.**  Persistent homology (PH) is a tool which summarizes the multi-scale topological features of a dataset in an object called a *persistence diagram*. Such topological summaries have been applied in machine learning tasks (Hensel et al., 2021), such as image segmentation (Hu et al., 2019; Clough et al., 2022; Shit et al., 2021; Waibel et al., 2022), and graph learning (Horn et al., 2021; Ballester & Rieck, 2024). The standard way to quantify the topological differences between datasets is to compute the Wasserstein distance between their persistence diagrams. However, there are two difficulties in directly applying persistence-based methods in adversarial deep learning tasks.

1. **Scalability.** Persistent homology of a large point cloud is prohibitively expensive to compute[1]. Even worse, the persistent homology algorithm is highly nontrivial to parallelize. Modern PH packages are either pure CPU implementations (Bauer, 2021; Pérez et al., 2021) or use a CPU-GPU hybrid algorithm (Zhang et al., 2020).

2. **Smoothness.** Persistent homology is differentiable almost everywhere, which allows us to compute backpropagate through PH layers; in fact, stochastic subgradient descent is provably convergent with respect to persistence-based functions (Carriere et al., 2021). However, in adversarial tasks, where the loss function is constantly changing, discontinuities in the gradient leads to highly unstable training dynamics (Wiatrak et al., 2019).

**Contributions.**  We address these two issues by modifying the two central parts of the classical persistence pipeline: the topological summary itself, as well as the metric used to compare them.

- **Topological Summary: Principal Persistence Measures.** To reduce the computational cost, we compute the persistent homology of many small batches of subsamples in parallel. By choosing a specific number of points depending on the homology dimension, the persistence computation significantly simplifies, and we obtain an object called the *principal persistence measure (PPM)* (Gómez & Mémoli, 2024). We provide a pure GPU implementation of the PPM, which enables a scalable methodology to incorporate topological features in larger-scale ML tasks. Moreover, subsampling results in a smoother features (Solomon et al., 2021), resulting in more stable training behavior.

- **Topological Metric: Maximum Mean Discrepancy for PPMs.** The Wasserstein distance is the primary metric used to compare PPMs (Gómez & Mémoli, 2024). In practice, one often uses entropic regularization to lower the computational cost (Cuturi, 2013; Lacombe et al., 2018). Despite this, it is still computationally expensive, and we use maximum mean discrepancy (MMD) metrics to compare PPMs. This coincides with the *persistence weighted kernels* introduced in (Kusano et al., 2016) for persistence diagrams. Our main theoretical results deals with establishing this metric in PPM framework.

  - Theorem 1 builds characteristic kernels for PPMs from kernels on $\mathbb{R}^2$.
  - Theorem 2 shows that these MMD metrics induce the same topology as Wasserstein.
  - Theorem 3 shows that gradients with respect to this metric are continuous.

  Theorem 1 and Theorem 2 adapt results from (Kusano et al., 2016; Divol & Lacombe, 2021) to the setting of PPMs, while to the authors' knowledge, Theorem 3 is novel.

These theoretical results imply that we can use PPM-Reg as an alternative to computationally expensive Wasserstein (or Sinkhorn) metrics, which produces a stable gradient for training deeper networks. In particular, the proposed methods allow us to incorporate topological features into large-scale machine learning tasks in a stable manner (Papamarkou et al., 2024, Section 4.2). We

---

[1]The worst-case time complexity is $O(m^3)$, where $m$ is the number of simplices. Computing dimension $k$ persistent homology of a point cloud with $n$ points can have up to $m = O(n^{k+1})$ simplices. However, for practical data sets, the complexity is often much lower; see (Otter et al., 2017) for further discussion.

demonstrate this empirically in Section 6, where we provide extensive experiments to demonstrate the efficacy of PPM-Reg in the GAN framework.

**Related Work.** The application of persistent homology in machine learning has been enabled by theoretical studies into the differentiability properties of PH (Carriere et al., 2021; Leygonie et al., 2022), which have also been extended to the multiparameter setting (Scoccola et al., 2024). However, large-scale computation of PH remains a challenge, though recent work has considered computational strategies for optimization problems (Nigmetov & Morozov, 2024; Luo & Nelson, 2024). Our MMD metric for PPMs is also related to work on kernels for persistence diagrams Kusano et al. (2016) and linear representations of persistence diagrams Divol & Lacombe (2021); Divol & Polonik (2019).

Subsampling methods for PH of metric measure spaces was introduced in (Blumberg et al., 2014), and used to approximate PH for point clouds (Chazal et al., 2015; Cao & Monod, 2022; Stolz, 2023). Furthermore, distributed approaches for computing the true PH of point clouds have been proposed in (Yoon & Ghrist, 2020; Torras-Casas, 2023) via spectral sequence methods. More recently, (Solomon et al., 2021) used subsampling methods for topological function optimization, motivated by the same issues of computational cost and instability of gradients (Bendich et al., 2020), and (Solomon et al., 2022) showed that such distributed persistence methods interpolate between geometric and topological features based on the number of subsamples. The starting point of this article is (Gómez & Mémoli, 2024), which introduces principal persistence measures.

## 2 LATENT SPACE MATCHING IN GENERATIVE ADVERSARIAL NETWORKS

Our primary consideration is the latent space matching in generative adversarial networks (GANs). A GAN is an unsupervised training framework consisting of a generator $g_{\boldsymbol{\omega}} : \mathbb{R}^N \to \mathbb{R}^M$, a discriminator $d_{\boldsymbol{\theta}} : \mathbb{R}^M \to \mathbb{R}^L$ and a value function $\mathcal{V} : \mathcal{P}(\mathbb{R}^L) \times \mathcal{P}(\mathbb{R}^L) \to \mathbb{R}$ (Goodfellow et al., 2014). Consider a set of training data, such as a collection of images, which we view as a probability measure $\mu$ on $\mathbb{R}^M$, the *data space*. The *generator* $g_{\boldsymbol{\omega}}$ is parameterized by $\boldsymbol{\omega} \in \mathbb{R}^G$, and its goal is to map a given noise measure $\nu$ on $\mathbb{R}^N$ (the *noise space*) to $\mathbb{R}^M$ such that $g_{\boldsymbol{\omega}}(\nu)$ can be interpreted as novel examples of $\mu$. The *discriminator* $d_{\boldsymbol{\theta}}$, parametrized by $\boldsymbol{\theta} \in \mathbb{R}^D$, performs dimensionality reduction, sending the data space to the *latent space* $\mathbb{R}^L$. Finally, the *value function* is used to quantify the difference between the real data $\mu$ and the generated data $g_{\boldsymbol{\omega}}(\nu)$ by $\mathcal{V}(d_{\boldsymbol{\theta}}(\mu), d_{\boldsymbol{\theta}}(g_{\boldsymbol{\omega}}(\nu)))$.

The generator is optimized such that it minimizes the value function, while the goal of the discriminator is to maximize it. Training algorithms (Goodfellow et al., 2014; Arjovsky et al., 2017; Gulrajani et al., 2017) have been proposed to find an equilibrium of the minimax problem, given by

$$\max_{\boldsymbol{\theta}} \min_{\boldsymbol{\omega}} \mathcal{V}\left(d_{\boldsymbol{\theta}}(\mu), d_{\boldsymbol{\theta}}(g_{\boldsymbol{\omega}}(\nu))\right). \tag{1}$$

In practice, parameters in $d_{\boldsymbol{\theta}}$ and $g_{\boldsymbol{\omega}}$ are updated alternatively. Common value functions used are metrics between probability distributions such as the Wasserstein distance, which has better theoretical properties in solving the minimax problem with gradient descent (Arjovsky et al., 2017), and the Cramer distance (Bellemare et al., 2017), which has unbiased gradients with mini-batch training. However, these metrics do not explicitly take topological features of the distributions into consideration. This motivates the use of a *topological regularizer*, which explicitly accounts for topological features in a non-equilibrium state. In particular, we consider value functions of the form

$$\mathcal{V} = \mathcal{L} + \lambda \mathcal{T}, \tag{2}$$

where $\mathcal{L}$ is the main loss function, $\lambda > 0$ is a hyperparameter, $\mathcal{T}$ is our proposed topological regularizer which will be introduced in the following sections.

## 3 PRINCIPAL PERSISTENCE MEASURES

We provide a streamlined exposition of the notion of principal persistence measures (Gómez & Mémoli, 2024). As there already exists several excellent references for persistent homology, we refer the reader to (Edelsbrunner & Harer, 2010; Dey & Wang, 2022; Hensel et al., 2021) for further background. Furthermore, we highlight the fact that we only consider persistent homology for simple point clouds, with an explicit definition in Equation (4). For a topological space $\mathcal{X}$, we use $\mathcal{P}(\mathcal{X})$ (resp. $\mathcal{P}_c(\mathcal{X})$) to denote the Borel probability measure (resp. with compact support) on $\mathcal{X}$.

Throughout this article, we consider persistent homology of point clouds with the Vietoris-Rips filtration.

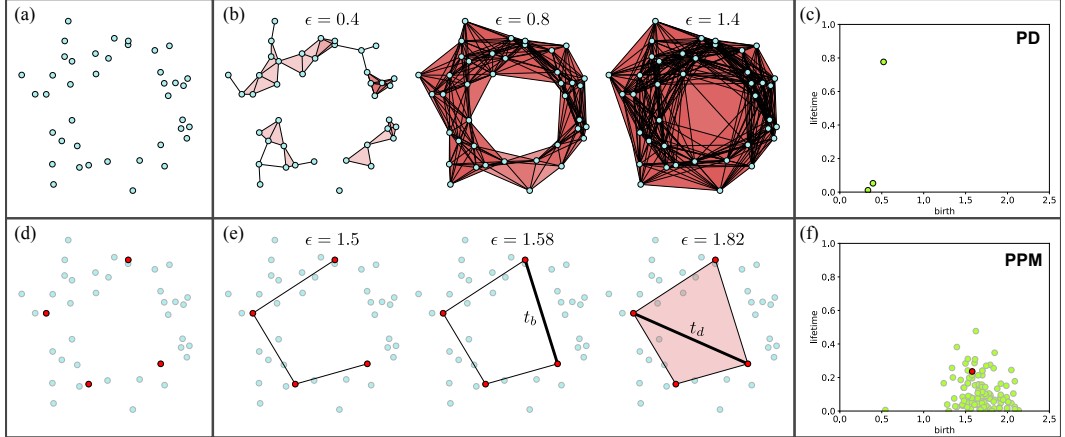

Figure 1: An illustration of PH and PPMs. (a) An example point cloud $X$. (b) Snapshots of the Vietoris-Rips filtration $X_\epsilon$ of $X$ at various $\epsilon$. Edges are added between $x_i$ and $x_j$ when $d(x_i, x_j) > \epsilon$ and higher simplices are added when all pairwise distances are greater than $\epsilon$. (c) The dimension 1 persistence diagram of $X$ in birth-lifetime coordinates. The one point with large lifetime represents the fact that there is a hole in the dataset which persists through multiple scales. (d) An example of a subsampling (in red) of $4 = 2q + 2$ points when $q = 1$. (e) Snapshots of the Vietoris-Rips filtration of the subsample, where the distances of the bold lines are $t_b$ and $t_d$. (f) The dimension 1 principal persistence measure of $X$, where the point given by the example subsample is shown in red.

**Persistent Homology.**     Let $X = \{x_i\}_{i=1}^N$, where $x_i \in \mathbb{R}^n$. *Persistent homology* (Edelsbrunner & Harer, 2010) of dimension $q \in \mathbb{N}$ builds a multi-scale topological summary of $X$ in three steps:

1. Construct a sequence of topological spaces $X_\epsilon$ representing the point cloud at a scale parameter $\epsilon > 0$, equipped with inclusion maps $X_\epsilon \hookrightarrow X_{\epsilon'}$ for $\epsilon < \epsilon'$ (see Figure 1(b)).

2. Compute the *dimension $q$ homology* of $X_\epsilon$ to obtain topological properties at each scale.

3. Track the *birth*, $b$, and *lifetime*, $\ell$, of topological features across scales by using the induced maps $H_q(X_\epsilon) \to H_q(X_{\epsilon'})$, and summarize this information as a multi-set $\mathrm{PH}_q(X) = \{(b_i, \ell_i)\}_{i=1}^r$ called a *persistence diagram*[2] (see Figure 1(c)).

The points $(b, \ell) \in \mathrm{PH}_q(X)$ in a persistence diagram are valued in the quotient of the half plane

$$\Omega := \{(b, \ell) \in \mathbb{R}^2 : \ell \geq 0\}/\{\ell = 0\}, \tag{3}$$

as topological features $(b, \ell) \in \mathrm{PH}_q(X)$ where points with trivial lifetime $\ell = 0$ are equivalent to the feature not existing. We view this as a pointed quotient metric space $(\Omega, d, *)$, where $d$ is the quotient of the Euclidean metric on $\mathbb{R}^2$ and $*$ represents the collapsed point $\{\ell = 0\}$.

**Persistent Homology of Small Point Clouds.**     It is shown in (Gómez & Mémoli, 2024, Theorem 4.4) that $\mathrm{PH}_q$ of a point cloud $S$ with exactly $2q + 2$ points has at most a single topological feature, and can be explicitly computed as follows. Given a point $x \in S$, let $x^{(1)}, x^{(2)} \in S$ denote the points such that $d(x, x^{(1)}) \geq d(x, x^{(2)}) \geq d(x, a)$ for all $a \in S - \{x^{(1)}, x^{(2)}\}$. Then,

$$\mathrm{PH}_q(S) = \{(t_b, t_d - t_b)\}, \quad t_b := \max_{x \in S} d(x, x^{(2)}), \quad t_d := \min_{x \in S} d(x, x^{(1)}) \tag{4}$$

whenever $t_d \geq t_b$, and $\mathrm{PH}_q(S) = \{*\}$ otherwise (see Figure 1(e)). As we will exclusively consider $\mathrm{PH}_q$ of $2q + 2$ points, we will consider this as a map $\mathrm{PH}_q : (\mathbb{R}^n)^{2q+2} \to \Omega$. We emphasize that Equation (4) is a significant simplification of the full persistent homology computation (Otter et al., 2017), and is the key to parallelized computations discussed in Section 6.1.

---

[2]Note that birth-lifetime coordinates are a linear transformation of the more standard birth-death coordinates. We use lifetime coordinates to simplify the kernel expressions in the following section.

**Principal Persistence Measures.** Principal persistence measures (PPMs) of dimension $q$ (Gómez & Mémoli, 2024) contains $PH_q$ of all subsamples $S$ of a point cloud $X$ with exactly $|S| = 2q + 2$ points. More formally, we will consider the more general setting of probability measures on $\mathbb{R}^n$ with compact support rather than point clouds[3] on $\mathbb{R}^n$. Then, the *PPM of dimension $q$* is defined as

$$\text{PPM}_q : \mathcal{P}_c(\mathbb{R}^n) \to \mathcal{P}(\Omega), \quad \text{PPM}_q(\mu) := (\text{PH}_q)_* \mu^{\otimes(2q+2)} \tag{5}$$

where $\mu^{\otimes n}$ is the product measure on $(\mathbb{R}^n)^{2q+2}$, and $(\text{PH}_q)_*$ is the pushforward map. In other words, we take $2q + 2$ i.i.d. samples from $\mu$ and compute $\text{PH}_q$ on each collection to obtain a probability measure on $\Omega$ (see Figure 1(f)).

**Metrics and Stability.** Let $p \geq 1$, and let $W_p$ denote the $p$-Wasserstein metric on $\mathbb{R}^n$ and $\Omega$. A key property shown in (Gómez & Mémoli, 2024, Theorem 3.8, Theorem 4.11) is that PPMs are *stable*:

$$W_p(\text{PPM}_q(\mu), \text{PPM}_q(\nu)) \leq C_q W_p(\mu, \nu), \quad \text{for all} \quad \mu, \nu \in \mathcal{P}_c(\mathbb{R}^n) \tag{6}$$

where $C_q > 0$ is a constant which depends on $q$. In particular, $p$-Wasserstein metrics on $\Omega$ for PPMs is the analogue of the partial $p$-Wasserstein distance for persistence diagrams.

## 4 MAXIMUM MEAN DISCREPANCY FOR PPMS

In order to further reduce the computational cost and obtain smoothness properties, we will use maximum mean discrepancy (MMD) metrics to compare PPMs. The kernels defined here adapted from the *persistence weighted kernels* of Kusano et al. (2016). We assume basic familiarity with kernels and refer the reader to Appendix A for background.

**Bounded PPMs and Notation.** Throughout this section, we work with *bounded* PPMs valued in

$$\Omega_T := \{(b, \ell) \in [0, T]^2 : \ell \geq 0\} / \{\ell = 0\} \tag{7}$$

for some $T > 0$. We continue to denote the collapsed point by $*$. Note that for $\mu \in \mathcal{P}_c(\mathbb{R}^n)$, where the support of $\mu$ has diameter $T$, we have $\text{PPM}_k(\mu) \in \mathcal{P}(\Omega_T)$. In order to simplify notation, we use $\Omega = \Omega_T$ throughout this section. We use the notation $z = (b, \ell)$ for elements in both $[0, T]^2$ and $\Omega$.

**Kernels on $\Omega$.** Following the construction in Kusano et al. (2016), we introduce a procedure to turn a kernel $k$ on $[0, T]^2$ into a kernel on $\Omega$. Suppose $k : [0, T]^2 \times [0, T]^2 \to \mathbb{R}$ is a kernel, where $\mathcal{H}$ is its reproducing kernel Hilbert space (RKHS), and let $\Phi : [0, T]^2 \to \mathcal{H}$ be the associated feature map given by $\Phi(z) = k(z, \cdot)$. We define a feature map $\Phi_\Omega : \Omega \to \mathcal{H}$ into the same RKHS by

$$\Phi_\Omega(z) = \ell \cdot \Phi(z) = \ell \cdot k(z, \cdot) \text{ when } \ell > 0 \quad \text{and} \quad \Phi_\Omega(*) = 0. \tag{8}$$

Then, for $z_1, z_2 \in \Omega - \{*\}$, the associated kernel $k_\Omega : \Omega \times \Omega \to \mathbb{R}$, satisfies

$$k_\Omega(z_1, z_2) := \langle \Phi_\Omega(z_1), \Phi_\Omega(z_2) \rangle_\mathcal{H} = \ell_1 \cdot \ell_2 \cdot k(z_1, z_2) \tag{9}$$

by the reproducing kernel property of $\mathcal{H}$, and $k_\Omega(*, z) = k_\Omega(z, *) = 0$. Note that $k_\Omega$ is continuous on $\Omega \times \Omega$. We denote the RKHS of $k_\Omega$ by $\mathcal{H}_\Omega$, where we have an embedding $\mathcal{H}_\Omega \hookrightarrow \mathcal{H}$ by definition.

**Characteristic Kernels on $\Omega$.** Recall that a kernel $k : \mathcal{X} \times \mathcal{X} \to \mathbb{R}$ (with associated feature map $\Phi : \mathcal{X} \to \mathcal{H}$) is *characteristic with respect to probability measures* $\mathcal{P}(\mathcal{X})$ if the *kernel mean embedding*, also denoted by $\Phi : \mathcal{P}(\mathcal{X}) \to \mathcal{H}$,

$$\Phi(\mu) := \mathbb{E}_{x \sim \mu}[\Phi(x)], \tag{10}$$

is injective. The following result shows that if we start with a characteristic kernel on $[0, T]^2$, the above procedure yields a characteristic kernel on $\Omega$. This can be shown using similar methods as (Kusano et al., 2016, Section 3.1) in the current setting, but we provide an independent proof of a slightly stronger statement in Theorem 5 of Appendix B.

**Theorem 1.** *Let $k : [0, T]^2 \times [0, T]^2 \to \mathbb{R}$ be a kernel which is universal with respect to $C([0, T]^2)$ (or equivalently, characteristic with respect to $\mathcal{P}([0, T]^2)$). Then, $k_\Omega : \Omega \times \Omega \to \mathbb{R}$ is characteristic with respect to $\mathcal{P}(\Omega)$.*

---

[3]This is a special case of the metric measure spaces used in (Gómez & Mémoli, 2024). Furthermore, we can associate a point cloud $X = \{x_i\}_{i=1}^N \subset \mathbb{R}^n$, with the uniform probability measure $\mu_X$ on $X$.

**MMD for Principal Persistence Measures.** A characteristic kernel $k_\Omega$ on $\Omega$ induces a metric on $\mathcal{P}(\Omega)$ via the norm, called the *maximum mean discrepancy (MMD)*,

$$\text{MMD}_k(\nu_1, \nu_2) := \left\| \Phi(\nu_1) - \Phi(\nu_2) \right\|_{\mathcal{H}_\Omega}. \tag{11}$$

Let $\nu_1 = \frac{1}{N} \left( \sum_{i=1}^n \delta_{x_i} + (N-n)\delta_* \right)$ and $\nu_2 = \frac{1}{M} \left( \sum_{j=1}^m \delta_{y_j} + (M-m)\delta_* \right)$ be discrete measures in $\mathcal{P}(\Omega)$, with $n$ and $m$ nontrivial points $x_i, y_j \in \Omega - \{*\}$ respectively. The MMD is given by

$$\text{MMD}_k^2(\nu_1, \nu_2) = \frac{1}{N^2} \sum_{i,j=1}^n k_\Omega(x_i, x_j) - \frac{2}{NM} \sum_{i=1}^n \sum_{j=1}^m k_\Omega(x_i, y_j) + \frac{1}{M^2} \sum_{i,j=1}^m k_\Omega(y_i, y_j). \tag{12}$$

The normalization is with respect to the total (including $*$) numbers of points in $\nu_1$ and $\nu_2$, but we only compute kernels between nontrivial points since $k_\Omega(*, z) = k_\Omega(z, *) = 0$. This enables the use of computable MMD metrics for PPMs. While the stability property in Equation (6) may no longer hold, MMD metrics yield the same topology on the space of probability measures (see also (Kusano et al., 2016, Theorem 3.2) in the finite setting). While a related result in a different context is given in (Divol & Lacombe, 2021, Proposition 5.1), we provide an independent proof in Appendix C.

**Theorem 2.** *Let $k$ be a characteristic kernel on $\Omega$. The $p$-Wasserstein metric $W_p$ and the MMD metric $MMD_k$ induce the same topology on $\mathcal{P}(\Omega)$.*

**Remark 1.** *By viewing persistence diagrams as measures (Divol & Lacombe, 2021; Giusti & Lee, 2023; Bubenik & Elchesen, 2022), persistence diagrams can be viewed as elements in $\mathcal{M}_{lin}(\Omega)$, defined in Equation (30). This includes persistence diagrams with possibly infinite cardinality (with finite total persistence). While in Kusano et al. (2016), analogous kernels are defined for finite persistence diagrams, our results hold for* persistence measures *in $\mathcal{M}_{lin}(\Omega)$.*

## 5 TOPOLOGICAL REGULARIZATION WITH PPMS

In this section, we introduce our proposed topological regularizer based on computing the PPM of probability measures and comparing the PPMs using MMD. Let $k_\Omega$ be a characteristic kernel as defined in the previous section. Returning to the notation of Section 2, we define our dimension $q$ topological regularizer, PPM-Reg, on the latent space $\mathbb{R}^L$ by $\mathcal{T}_q : \mathcal{P}_c(\mathbb{R}^L) \times \mathcal{P}_c(\mathbb{R}^L) \to \mathbb{R}$ by

$$\mathcal{T}_q(\mu, \nu) := \text{MMD}_{k_\Omega}(\text{PPM}_q(\mu), \text{PPM}_q(\nu)) = \left\| \Phi_\Omega(\text{PPM}_q(\mu)) - \Phi_\Omega(\text{PPM}_q(\nu)) \right\|_{\mathcal{H}_\Omega}. \tag{13}$$

When we apply this in the GAN setting, we consider $\mathcal{T}_q\left( d_{\boldsymbol{\theta}}(\mu), d_{\boldsymbol{\theta}}(g_{\boldsymbol{\omega}}(\nu)) \right)$, where $\nu \in \mathcal{P}_c(\mathbb{R}^N)$ is the noise measure, and $\mu \in \mathcal{P}_c(\mathbb{R}^M)$ is the data measure. Let $\mathfrak{T}_q : \mathbb{R}^G \times \mathbb{R}^D \to \mathbb{R}$ be

$$\mathfrak{T}_q(\boldsymbol{\omega}, \boldsymbol{\theta}) := \mathcal{T}_q\left( d_{\boldsymbol{\theta}}(\mu), d_{\boldsymbol{\theta}}(g_{\boldsymbol{\omega}}(\nu)) \right). \tag{14}$$

Our main result of this section is to show that $\mathfrak{T}_q$ is smooth with respect to $\boldsymbol{\omega}$ and $\boldsymbol{\theta}$ given sufficient smoothness conditions on the underlying measures and the discriminator and generator. Recall that a function $f : \mathbb{R}^n \to \mathbb{R}^m$ is a $C^1$ function if all first derivatives of $f$ are continuous. The following is our main theoretical result, proved in Appendix D.

**Theorem 3.** *Let $k_\Omega$ be a characteristic kernel. Suppose $\mu \in \mathcal{P}_c(\mathbb{R}^M)$ and $\nu \in \mathcal{P}_c(\mathbb{R}^N)$ have $C^1$ densities. Suppose the joint functions $G : \mathbb{R}^G \times \mathbb{R}^N \to \mathbb{R}^M$ defined by $G(\boldsymbol{\omega}, x) = g_{\boldsymbol{\omega}}(x)$ and $D : \mathbb{R}^D \times \mathbb{R}^M \to \mathbb{R}^L$ defined by $D(\boldsymbol{\theta}, y) = d_{\boldsymbol{\theta}}(y)$ be $C^1$ functions. Then, $\mathfrak{T}_q$ is a $C^1$ function wherever the PPM is not the trivial measure at the origin.*

## 6 EXPERIMENTS AND RESULTS

We provide empirical experiments which demonstrates the efficacy of PPM-Reg as a topological regularizer. First, in Section 6.2, we provide an expository shape matching experiment to illustrate the behavior of PPM-Reg, and provide computational comparisons. Next, in Section 6.3, we apply PPM-Reg to a GAN-based generative modelling problem, consistently improving the *generative* quality of GANs. Finally, in Section 6.4, we consider a GAN-based semi-supervised learning problem, which demonstrates the effectiveness of PPM-Reg in improving the *discriminative* ability of GANs. Due to space limitations, we have placed implementation details and additional experiments for each of the three settings in Appendix E, Appendix F, and Appendix G.[4]

---

[4]Code & supp.: `https://github.com/htwong-ai/ScalableTopologicalRegularizers`.

### 6.1 COMPUTATIONAL SETUP AND IMPLEMENTATION OVERVIEW

**Cramer Distance.** The Wasserstein and Cramer distance are both probability metrics that are sensitive to the geometry of the change in distribution (Bellemare et al., 2017). Moreover, the Cramer distance does not depend on hyperparameters which simplifies our comparison. We primarily use the Cramer metric as our main loss function $\mathcal{L}$. Following the definition of (Bellemare et al., 2017), for $\mu, \nu \in \mathcal{P}(\mathbb{R}^d)$. The Cramer Distance $\mathcal{E}(\mu, \nu)$ is defined as

$$\mathcal{E}(\mu, \nu) \coloneqq \mathbb{E}_{x \sim \mu}\big[\mathcal{D}(x)\big] - \mathbb{E}_{y \sim \nu}\big[\mathcal{D}(y)\big], \quad \mathcal{D}(z) \coloneqq \mathbb{E}_{y' \sim \nu}\big[\|z - y'\|_2\big] - \mathbb{E}_{x' \sim \mu}\big[\|z - x'\|_2\big]$$

where $x, x'$ (resp. $y, y'$) are independent random variables with law $\mu$ (resp. $\nu$). We do not take the gradient estimation in (Bellemare et al., 2017, Appendix C.3) as we obtain sufficient samples.

**Implementation of PPM.** For all experiments, we use $s$ subsamples from $\mu^{\otimes(2q+2)}$ to approximate the PPM (using the same number of subsamples for dimension 0 and 1). The persistent homology of each subsample is computed using Equation (4) in parallel on the GPU. Throughout these experiments, our base kernel is the radial basis function (RBF) kernel $k_{\text{RBF}}(z_1, z_2) = \exp\big(-\|z_1 - z_2\|^2/2\sigma\big)$, where the width $\sigma > 0$ is a hyperparameter. Thus, the induced kernel $k_\Omega : \Omega \times \Omega \to \mathbb{R}$ from Equation (9) evaluated on $z_i = (b_i, \ell_i) \in \Omega$ is

$$k_\Omega(z_1, z_2) = \ell_1 \cdot \ell_2 \exp\big(-\|z_1 - z_2\|^2/2\sigma\big). \tag{15}$$

We use Equation (12) to compute the MMD metric between PPMs. Furthermore, we use a weighted combination of dimension 0 and 1 PPM in our topological regularizer, such that

$$\mathcal{T} = \lambda_0 \mathcal{T}_0 + \lambda_1 \mathcal{T}_1, \tag{16}$$

where the *weights* $\lambda_0, \lambda_1 > 0$ are hyperparameters. We will call this **PPM-Reg**.

### 6.2 SHAPE MATCHING

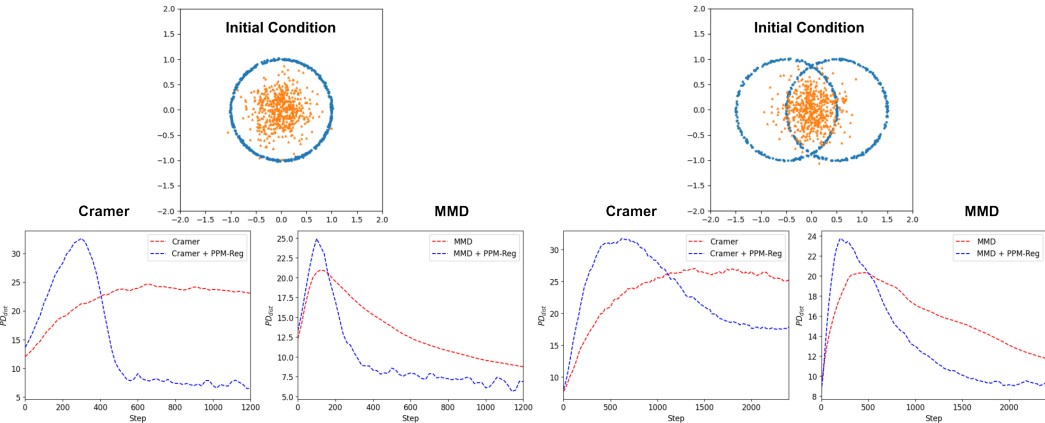

Figure 2: Visual example of PPM-Reg in a shape matching experiment using Cramer or MMD as the main loss function. 1st row: Plots of a reference point cloud (in blue) and the initial condition of a random point cloud (in orange). 2nd Row: Plots of 2-Wasserstein distance between 1-dimensional persistent homology between the reference shape and training shape over optimization steps.

Our task is to optimize the individual points of a point cloud to match the "shape" of a reference point cloud using a loss function of the form $\mathcal{L} + \mathcal{T}$, which operates directly on the ambient space of the point clouds. We choose $\mathcal{L}$ to either be the Cramer distance (Bellemare et al., 2017) or an MMD metric using an RBF kernel (with width $\sigma = 0.1$). Our aim in this expository experiment is twofold.

1. Demonstrate the ability of PPM-Reg to regularize *topological* features in point clouds by comparing the true (non-subsampled) persistence diagrams of the trained and reference shapes. Our focus is on showing that this occurs near the beginning of the optimization, since in GAN settings, the regularization is important away from global minima (see Introduction).
2. Show the computational efficiency of PPM-Reg, which enables its use in later experiments.

**Shape Matching Experiment.** Our main results are summarized in Figure 2. We choose two reference shapes in $\mathbb{R}^2$ for visualization purposes: a circle, and the union of two intersecting circles. In the second row of Figure 2, we plot the 2-Wasserstein distance between the 1-dimensional full (non-subsampled) persistence diagrams between the fixed reference and the trained point cloud, as a function of optimization steps. In each case, we see that adding PPM-Reg significantly reduces the topological distance. An interesting feature of each of these plots is that there is an initial spike in the PD distance when using PPM-Reg. Empirically, this is due to the fact that the point cloud must first move through a regime with trivial topological structure before PPM-Reg can faithfully match the topology of the reference. The training behavior is best understood by observing the dynamics of the optimization, and we provide animations of these experiments in the supplementary material.

**Computational Comparisons.** The efficiency of the PPM-Reg is derived from two major components: the parallelizable PPM and the iterative-free MMD. Table 1 empirically shows the computational benefit of each component as the size of the point cloud and the number of subsamples $s$ are varied. We use Cramer as the main loss, and consider the computational cost of using PPM-Reg, W-PPM-Reg and PD-Reg. **PD-Reg** computes the 2-Wasserstein distance between dimension 0 and 1 full persistent homology with Vietoris-Rips filtration. **W-PPM-Reg** computes the 2-Wasserstein distance between PPMs of dimension 0 and 1. We use the `torch-topological` package (Lab) to compute persistent homology and Wasserstein distances.

Our aim is to compare the real-world usage of these methods using the circle experiment. Thus, PPM-Reg computations are performed on a GPU, while PD-Reg and W-PPM-Reg uses hybrid CPU-GPU methods. The computational cost of PD-Reg grows exponentially with respect to the size of the point cloud. While the computational cost of W-PPM-Reg is sublinear with respect to the size of the point cloud, the cost is exponential with respect to the number of subsamples $s$. Remarkably, due to parallelization, PPM-Reg is sublinear with respect to the number of subsamples $s$ and is nearly constant as the size of the point cloud increases. In the following experiments, we find that $s = 1024$ and $s = 2048$ performs well in practice. In summary, using MMD mediates the drawback of the increased number of features extracted by PPM, resulting in our significantly faster PPM-Reg. With parallelization, our pure `PyTorch` implementation of PPM-Reg outperforms highly optimized low-level CPU implementations used in `torch-topological`.

| | Cramer + PPM-Reg | | | Cramer + W-PPM-Reg | | | Cramer + PD-Reg |
|---|---|---|---|---|---|---|---|
| No. points | $s = 512$ | $s = 1024$ | $s = 2048$ | $s = 512$ | $s = 1024$ | $s = 2048$ | |
| 128 | $0.55 \pm 0.005$ | $0.61 \pm 0.007$ | $0.98 \pm 0.004$ | $3.06 \pm 0.033$ | $13.28 \pm 0.198$ | $73.00 \pm 3.323$ | $1.99 \pm 0.048$ |
| 256 | $0.56 \pm 0.008$ | $0.61 \pm 0.005$ | $0.99 \pm 0.005$ | $3.25 \pm 0.083$ | $14.04 \pm 0.187$ | $80.11 \pm 2.545$ | $10.43 \pm 0.075$ |
| 512 | $0.56 \pm 0.010$ | $0.61 \pm 0.014$ | $0.98 \pm 0.006$ | $3.43 \pm 0.111$ | $16.43 \pm 0.458$ | $91.29 \pm 3.081$ | $107.11 \pm 2.837$ |
| 1024 | $0.57 \pm 0.005$ | $0.61 \pm 0.005$ | $1.00 \pm 0.007$ | $3.90 \pm 0.092$ | $19.45 \pm 0.468$ | $121.76 \pm 3.424$ | $655.58 \pm 11.823$ |

Table 1: Running time of 100 gradient steps (in seconds) in matching circle form randomly initialize gaussian in $\mathbb{R}^2$ with GPU computation enable. The averages are computed over 10 runs.

**Imperfect Convergence.** In Appendix E.2, we observe the same trends in additional modified experiments, which prevent the centroid of the trained shape from converging to the centroid of the reference. This is done to mimic the GAN setting where training algorithms often converge to saddle points rather than global minima (Berard et al., 2019; Liang & Stokes, 2019).

## 6.3 UNCONDITIONAL IMAGE GENERATION

Next, we consider the use of PPM-Reg in an unconditional image generation task, which is the standard benchmark to evaluate GANs (Goodfellow et al., 2014; Arjovsky et al., 2017).

**Network Architecture and Implementation Details.** We use a ResNet based CNN as the generator $g_{\omega}$, which takes a 128-dimensional noise vector as input. We use a CNN as the discriminator $d_{\theta}$ and the output of the network is a 128-dimensional latent vector. We compare the Cramer value function $\mathcal{V} = \mathcal{L}$, with the use of PPM-Reg $\mathcal{V} = \mathcal{L} + \mathcal{T}$. As our network differs from (Bellemare et al., 2017), we retrain both regularized and unregularized networks for a fair comparison.

**Dataset and Evaluation Metrics.** We consider the CelebA (Liu et al., 2015) and Anime-Face (Churchill & Chao, 2019) datasets. Images are centered and resized to $32 \times 32$. While the Frechét Inception Distance (FID) (Heusel et al., 2017) is a popular metric to evaluate the distance between generated and real images, recent empirical work has thoroughly investigated several drawbacks of FID (Horak et al., 2021; Stein et al., 2024; Jayasumana et al., 2024). Instead, we adopt three metrics:

1. CMMD (Jayasumana et al., 2024): MMD of CLIP embeddings (Radford et al., 2021),
2. $FD_{Dinov2}$ (Stein et al., 2024): Frechét Distance of Dinov2 (Oquab et al., 2024) embeddings
3. $WD_{latent}$: 2-Wasserstein distance of CLIP embeddings (Radford et al., 2021)

In Table 2, these are computed by sampling / generating 10K images from the data set / network. We report CMMD, $FD_{Dinov2}$ and $WD_{latent}$ using the epoch with the smallest CMMD.

| | AnimeFace | | | CelebA | | |
|---|---|---|---|---|---|---|
| | CMMD | $FD_{Dinov2}$ | $WD_{latent}$ | CMMD | $FD_{Dinov2}$ | $WD_{latent}$ |
| Cramer (Bellemare et al., 2017) | 0.73 | 953.99 | 0.6294 | 0.72 | 722.86 | 0.6795 |
| Cramer + PPM-Reg | 0.56 | 780.68 | 0.6080 | 0.58 | 700.73 | 0.6666 |

Table 2: Quantitative evaluation on $32 \times 32$ image generation, values are reported at the epoch with the smallest CMMD.

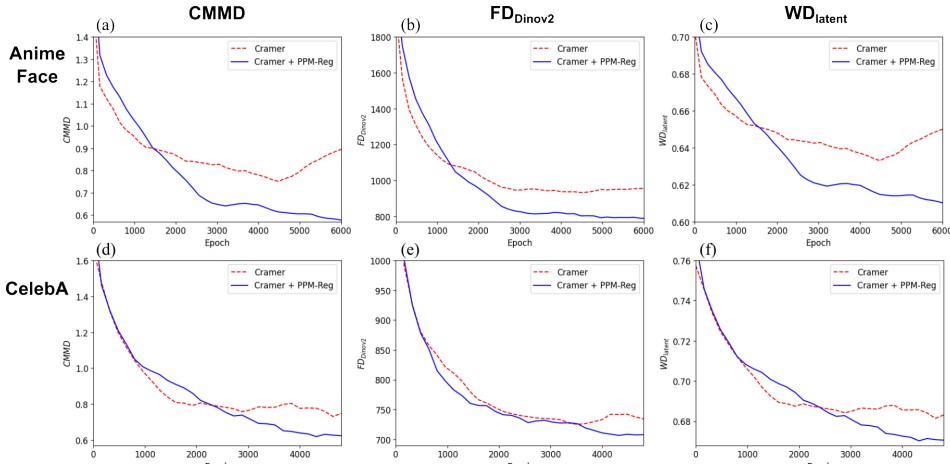

Figure 3: CMMD (a,d), $FD_{Dinov2}$ (b,e) and $WD_{latent}$ (c,f) versus training epochs for the AnimeFace (a-c) and CelebA (d-f) dataset. 10K samples are randomly generated to compute distances; moving averages with a window of 5 are used to smooth the values. Distances recorded every 160 epochs.

**Results.** Figure 3 tracks these three metrics during training. Figure 3 (a,b,d,e) shows that using PPM-Reg improves image generation quality for both AnimeFace and CelebA. The Wasserstein distance can better detect geometric information in embedding space. Tracking $WD_{latent}$ in Figure 3 (c,f) shows that adding PPM-Reg provides more information and helps discover geometric structures in the latent space in an unsupervised way. This reinforces work that shows that persistence-based methods are able to effectively measure image generation quality (Zhou et al., 2021; Khrulkov & Oseledets, 2018; Barannikov et al., 2021; Charlier et al., 2019). As training progresses, improved Cramer loss does not always lead to improved evaluation metrics (Figure 3 (a,c)). Our reported results use CMMD as an early stopping criterion which is prohibitively expensive to compute in practice. In contrast, the evaluation metrics consistently decrease with respect to training time, and this implies that may not need to compute additional metrics for early stopping. In Appendix F.2, we consider larger ($64 \times 64$) image generation experiments with the CelebA and LSUN Kitchen datasets, and find similarly improved results, demonstrating the efficacy of PPM-Reg in larger-scale experiments.

## 6.4 SEMI-SUPERVISED LEARNING

Semi-supervised learning (SSL) methods use unlabeled data alongside a small amount of labeled data to train a classification network (Yang et al., 2022). SSL often assumes that classification problems are supported on low-dimensional manifolds, which allows a network to learn the classification problem with limited labels (Niyogi, 2013). In practice, knowledge of the low-dimensional manifold can be learned by encoding the unlabeled data to latent representations. With few labeled data points, a simple classifier is trained using those latent representations (Wang et al., 2021a; Decourt & Duong, 2020; Truong et al., 2019; Das et al., 2021). Here, we demonstrate PPM-Reg can help encode more informative latent representations, and significantly reduce classification error in SSL.

**Network Architecture and Implementation Details.** We use a deconvolutional network as $g_{\omega}$, which takes a 64 dimension noise vector as input. We use a CNN as $d_{\theta}$ and the output of the network is a 64 dimension latent vector. We use an MLP as a classifier parameterized by $\gamma$, termed $c_{\gamma}$. We first learn the latent representations using a Cramer GAN (Bellemare et al., 2017) framework with all available data, and compare it against the addition of PPM-Reg. After training the GAN, the discriminator $d_{\theta}$ is frozen and its output is used as the features to train the classifier $c_{\gamma}$ with the subset of training samples. For a comparison without latent representations learning, we consider a "Baseline", where $d_{\theta}$ and $c_{\gamma}$ are trained together as a classifier (without the generative part).

**Dataset and Evaluation Metrics.** We compare the SSL performance with Fashion-MNIST (Xiao, 2017), Kuzushiji-MNIST (Clanuwat et al., 2018) and MNIST. In these experiments, 200 and 400 labels are randomly sampled from the data set. Due to the inherent randomness in sampling few labels, experiments are repeated ten times and the statistics of the best test-set accuracy are reported.

| | Fashion-MNIST | | Kuzushiji-MNIST | | MNIST | |
|---|---|---|---|---|---|---|
| Number of labels | 200 | 400 | 200 | 400 | 200 | 400 |
| Baseline | $67.18 \pm 0.95$ | $71.00 \pm 0.83$ | $48.40 \pm 1.79$ | $55.10 \pm 1.55$ | $80.52 \pm 1.49$ | $86.39 \pm 1.14$ |
| Cramer | $62.70 \pm 1.25$ | $68.58 \pm 1.08$ | $47.77 \pm 1.40$ | $56.06 \pm 1.88$ | $71.10 \pm 1.52$ | $78.26 \pm 1.30$ |
| Cramer + PPM-Reg | $\mathbf{76.84 \pm 1.23}$ | $\mathbf{80.59 \pm 0.69}$ | $\mathbf{75.78 \pm 1.99}$ | $\mathbf{79.33 \pm 1.69}$ | $\mathbf{96.62 \pm 0.39}$ | $\mathbf{97.33 \pm 0.21}$ |

Table 3: Test-set classification accuracy (%) on Fashion-MNIST, Kuzushiji-MNIST and MNIST. with 200 and 400 labeled examples. The average and the error bar are computed over 10 runs.

**Result.** Table 3 shows the test classification accuracy, where we use only 0.33% (200) and 0.66% (400) of the total number of labels. Compared with Baseline, only using Cramer does not significantly improve the classification accuracy in SSL. Note that while the Cramer GAN (without PPM-Reg) has reasonable generative ability, shown in Figure 7, this does not imply strong discriminator performance in SSL. Remarkably, using PPM-Reg significantly improves the classification accuracy. For example, compared with the Baseline, Kuzushiji-MNIST has gain 27.38% improvement with 200 labels. Notably, with the latent representations learned with PPM-Reg, we can get a good accuracy using only 0.66% of the labels. This section demonstrates that discovering topological structures in latent space is not only useful in generative tasks, but can be leveraged to massively improve classification accuracy when very few labels are available. In Appendix G.2, we observe similar performance gains in additional experiments on the SVHN dataset.

## 7 CONCLUSION

In this article, we propose a novel method for stable and scalable topological regularization based on the subsampling principle of PPMs, opening up the possibility of detecting topological information in larger-scale machine learning problems. We introduced a theoretical framework for using kernel methods and MMD metrics for PPMs, and demonstrated the efficacy of this methodology in a variety of experimental settings. This work suggests several directions for future study. From the theoretical and computational perspective, can we develop parallelizable approximate computations in more general settings? From an applied perspective, how can we leverage approximate topological summaries in further machine learning tasks such as classification or regression?

ACKNOWLEDGMENTS

This work was supported by the Hong Kong Innovation and Technology Commission (InnoHK Project CIMDA).

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

## A  BACKGROUND ON KERNELS AND MMD

In this section, we provide a brief overview of kernel methods, leading towards the maximum mean discrepancy. For further background, we refer the reader to (Muandet et al., 2017).

**Kernels and Feature Maps**    Suppose $\mathcal{X}$ is a topological space on which we wish to study either functions $f : \mathcal{X} \to \mathbb{R}$ or measures $\mu \in \mathcal{P}(\mathcal{X})$. A kcommon way to consider such objects is by using a *feature map*

$$\Phi : \mathcal{X} \to \mathcal{H} \tag{17}$$

into some Hilbert space $\mathcal{H}$. Heuristically, we can consider

- **functions on** $\mathcal{X}$ via linear functionals $\langle \ell, \Phi(\cdot) \rangle_{\mathcal{H}} : \mathcal{X} \to \mathbb{R}$ where $\ell \in \mathcal{H}$, and
- **measures on** $\mathcal{X}$ by considering the *kernel mean embedding* (which we also denote by $\Phi$), defined by

$$\Phi : \mathcal{P}(\mathcal{X}) \to \mathcal{H}, \quad \Phi(\mu) = \int_{\mathcal{X}} \Phi(x) d\mu(x). \tag{18}$$

Given this feature map, we can define a positive-definite *kernel* $k : \mathcal{X} \times \mathcal{X} \to \mathbb{R}$ defined by

$$k(x, y) := \langle \Phi(x), \Phi(y) \rangle_{\mathcal{H}}. \tag{19}$$

**Reproducing Kernel Hilbert Spaces.**    In fact, we can also go in the other direction and start with a continuous positive definite kernel $k : \mathcal{X} \times \mathcal{X} \to \mathbb{R}$, and obtain a feature map from $\mathcal{X}$ into a Hilbert space of functions. In particular, we define $\mathcal{H}$ to be the completion of the linear span of functions $\{k(x, \cdot) : \mathcal{X} \to \mathbb{R} : x \in \mathcal{X}\}$, equipped with the inner product

$$\langle k(x, \cdot), k(y, \cdot) \rangle := k(x, y). \tag{20}$$

By the Moore-Aronszajn theorem (Aronszajn, 1950), $\mathcal{H}$ is a Hilbert space with the *reproducing kernel* property: for any $f \in \mathcal{H}$ and $x \in \mathcal{X}$, we have

$$\langle f, k(x, \cdot) \rangle = f(x). \tag{21}$$

Thus, $\mathcal{H}$ is a *reproducing kernel Hilbert space (RKHS)*. Note that this is a Hilbert space of functions, $\mathcal{H} \subset C(\mathcal{X}, \mathbb{R})$. Then, we can define a feature map $\Phi : \mathcal{X} \to \mathcal{H}$ by

$$\Phi(x) := k(x, \cdot). \tag{22}$$

**Universality and Characteristicness**    We wish to consider feature maps (or kernels) which satisfy additional properties such that they can approximate functions and characterize measures. Let $\mathcal{F} \subset C(\mathcal{X}, \mathbb{R})$ be a topological vector space, and suppose $\mathcal{M}$ is a space of measures on $\mathcal{X}$. A feature map $\Phi : \mathcal{X} \to \mathcal{H}$ (with associated kernel $k : \mathcal{X} \times \mathcal{X} \to \mathbb{R}$), where $\mathcal{H} \subset \mathcal{F}$ is (Simon-Gabriel & Schölkopf, 2018)

- **universal** with respect to $\mathcal{F}$ if $\mathcal{H}$ is dense in $\mathcal{F}$ (we can approximate functions in $\mathcal{F}$ using functions in $\mathcal{H}$); and
- **characteristic** with respect to $\mathcal{M}$ if the kernel mean embedding in Equation (18) is injective.

**Maximum Mean Discrepancy**    Given a feature map $\Phi : \mathcal{X} \to \mathcal{H}$ (with kernel $k$) characteristic to the space of probability measures $\mathcal{P}(\mathcal{X})$, we can use the Hilbert space norm to define a metric on this space of measures. In fact, this is equivalent to the notion of *maximum mean discrepancy (MMD)* from statistics. In particular, given a function class $\mathcal{F} \subset C(\mathcal{X}, \mathbb{R})$, we define the MMD with respect to $\mathcal{F}$ by

$$\mathrm{MMD}_{\mathcal{F}}(\mu, \nu) := \sup_{f \in \mathcal{F}} \left( \mathbb{E}_{x \sim \mu}[f(x)] - \mathbb{E}_{y \sim \nu}[f(y)] \right). \tag{23}$$

Now, by (Gretton et al., 2012, Lemma 4), if we choose $\mathcal{F}$ to be the unit ball of the RKHS $\mathcal{H}$ (with respect to a characteristic kernel $k$), the MMD with respect to $\mathcal{F}$ is exactly the Hilbert space norm,

$$\mathrm{MMD}_{\mathcal{F}}(\mu, \nu) = \|\Phi(\mu) - \Phi(\nu)\|_{\mathcal{H}} =: \mathrm{MMD}_k(\mu, \nu), \tag{24}$$

where the right hand side is how we define $\mathrm{MMD}_k$ in Equation (11).

## B  CHARACTERISTIC KERNELS ON $\Omega$.

In this appendix, we provide a detailed discussion of the proof of Theorem 1.

We begin by characterizing some of the elements in the RKHS $\mathcal{H}_\Omega$.

**Lemma 1.** *The RKHS $\mathcal{H}_\Omega$ satisfies*

$$\mathcal{H}_\Omega \supset \{g : \Omega \to \mathbb{R} \,:\, g(b, \ell) = \ell \cdot f(b, \ell), \, f \in \mathcal{H}\}. \tag{25}$$

*Proof.* By Moore-Aronszajn (Aronszajn, 1950), an element of $g \in \mathcal{H}_\Omega$ is defined by a convergent series

$$g(b, \ell) = \sum_{i=1}^{\infty} c_i k_\Omega((b_i, \ell_i), (b, \ell)) = \ell \sum_{i=1}^{\infty} c_i \ell_i k((b_i, \ell_i), (b, \ell)) = \ell \cdot f(b, \ell), \tag{26}$$

where $f(b, \ell) = \sum_{i=1}^{\infty} c_i \ell_i k((b_i, \ell_i), (b, \ell))$. Note that if the coefficient series for $f$ given by $\sum_{i=1}^{\infty} |c_i| \ell_i k((b_i, \ell_i), (b_i, \ell_i))$ is convergent, then the coefficient series for $g$ given by

$$\sum_{i=1}^{\infty} |c_i| k_\Omega((b_i, \ell_i), (b_i, \ell_i)) = \sum_{i=1}^{\infty} |c_i| \ell_i^2 k((b_i, \ell_i), (b_i, \ell_i)) \le T \sum_{i=1}^{\infty} |c_i| \ell_i k((b_i, \ell_i), (b_i, \ell_i)) \tag{27}$$

is also convergent, since $\ell_i \le T$. $\qquad\square$

**Universality and Characteristicness.** Our first main result concerns the transfer of universal and characteristic properties from $k$ to $k_\Omega$. First, we define the space of *linear-growth continuous functions on $\Omega_T$* by

$$C_{\mathrm{lin}}(\Omega_T) := \{\ell \cdot f(b, \ell) \in C(\Omega_T) \,:\, f(b, \ell) \in C([0, T]^2)\}, \tag{28}$$

where we equip it with the norm defined on $g = \ell \cdot f$ by

$$\|g\|_{\mathrm{lin}} := |f|_\infty. \tag{29}$$

Note that for all $g \in C_{\mathrm{lin}}(\Omega)$ have the property that $g(*) = 0$, and $(C_{\mathrm{lin}}(\Omega_T), \|\cdot\|_{\mathrm{lin}})$ and $(C([0, T]^2), |\cdot|_\infty)$ are isometric Banach spaces.

**Theorem 4.** *Let $k : [0, T]^2 \times [0, T]^2 \to \mathbb{R}$ be a kernel on $[0, T]^2$ universal to $C([0, T]^2)$. Then, $k_\Omega : \Omega \times \Omega \to \mathbb{R}$ is universal with respect to $C_{lin}(\Omega_T)$.*

*Proof.* Let $g \in C_{\mathrm{lin}}(\Omega_T)$ and suppose $g = \ell \cdot f$ for some $f \in C([0, T]^2)$. Because $k$ is universal, there exists $f_n \in \mathcal{H}$ such that $|f_n - f|_\infty \to 0$. Then, by Lemma 1, $g_n = \ell \cdot f_n \in \mathcal{H}_\Omega$, and furthermore, by the definition of $\|\cdot\|_{\mathrm{lin}}$ in Equation (29), we have $\|g_n - g\|_{\mathrm{lin}} \to 0$. Thus, $k_\Omega$ is universal with respect to $C_{\mathrm{lin}}(\Omega_T)$. $\qquad\square$

The main result we wish to obtain is characteristicness with respect to measures on $\Omega$. We begin by applying the duality theorem of (Simon-Gabriel & Schölkopf, 2018, Theorem 6) which immediately implies characteristicness with respect to the topological dual of $C_{\mathrm{lin}}(\Omega)$, which we equip with the weak-* topology with respect to $C_{\mathrm{lin}}(\Omega)$.

**Corollary 1.** *Let $k : [0, T]^2 \times [0, T]^2 \to \mathbb{R}$ be a kernel on $[0, T]^2$ universal to $C([0, T]^2)$. Then, $k_\Omega$ is characteristic with respect to $C_{lin}(\Omega)^*$.*

Our next task is to show that our desired measures are contained in this dual space.

**Theorem 5.** *Let $q : [0, T]^2 \to \Omega$ denote the quotient map, and let $s : [0, T]^2 \to [0, \infty)$ be defined by $s(b, \ell) = \ell$ for $\ell > 0$ and $s(b, 0) = 0$. Define*

$$\mathcal{M}_{lin}(\Omega) := \left\{ q_* \mu \in \mathcal{M}(\Omega) \,:\, \mu \in \mathcal{M}([0, T]^2), \, \mu(\{\ell = 0\}) = 0, \, \left| \int_{[0,T]^2} \ell d\mu \right| < \infty \right\}, \tag{30}$$

*then $\mathcal{M}_{lin}(\Omega) \subset C_{lin}(\Omega)^*$.*

*Proof.* Let $q_*\mu \in \mathcal{M}_{\text{lin}}(\Omega)$. Then, for $g = \ell \cdot f \in C_{\text{lin}}(\Omega)$, we have

$$\left| \int_\Omega \ell \cdot f(b,\ell) q_* d\nu \right| = \left| \int_{[0,T]^2} \ell \cdot f(b,\ell) d\mu \right| \leq \|g\|_{\text{lin}} \left| \int_{[0,T]^2} \ell d\mu \right|, \tag{31}$$

where we use $\mu(\{\ell = 0\}) = 0$ in the first equality. Thus, $q_*\mu$ is a bounded (hence continuous) linear functional. $\square$

As we wish to use this kernel to study PPMs, we are interested in probability measures on $\Omega$. However, the elements in the dual only contain measures which are trivial on $* \in \Omega$ (whereas PPMs may have nontrivial mass on $*$), and thus we first define a different representation of probability measures. In particular, we define

$$\mathcal{P}_{\text{lin}}(\Omega) := \{\nu \in \mathcal{M}_{\text{lin}}(\Omega) : |\nu| \leq 1\} = \{\nu \in \mathcal{M}(\Omega) : \nu(\{*\}) = 0, |\nu| \leq 1\} \subset \mathcal{M}_{\text{lin}}(\Omega). \tag{32}$$

Note that the equality holds since the moment condition in Equation (30) is immediately satisfied since the measures are finite with bounded support. With the following two lemmas, we show that this coincides with $\mathcal{P}(\Omega)$.

**Lemma 2.** *The space $C_{lin}(\Omega)$ is dense in $C_0(\Omega)$ equipped with the uniform topology.*

*Proof.* Let $g \in C_0(\Omega)$, and since $g$ is uniformly continuous (since $\Omega$ is compact) and $g(*) = 0$, for every $\epsilon > 0$, there exists some $\ell_\epsilon$ such that $g(b,\ell) < \epsilon$ whenever $\ell < \ell_\epsilon$. Now, define $f_n \in C([0,T]^2)$ by

$$f_n(b,\ell) = \frac{g(b,\ell)}{\ell} \quad \text{for} \quad \ell \geq \ell_{1/n} \quad \text{and} \quad f_n(b,\ell) = \frac{g(b,\ell_{1/n})}{\ell_{1/n}} \quad \text{for} \quad \ell < \ell_{1/n}. \tag{33}$$

Then, define $g_n(b,\ell) = \ell \cdot f(b,\ell)$, where $g_n(b,\ell) = g(b,\ell)$ whenever $\ell \geq \ell_{1/n}$. When $\ell < \ell_{1/n}$, we have

$$|g_n(b,\ell) - g(b,\ell)| = |g_n(b,\ell_{1/n} - g(b,\ell)| \leq \frac{2}{n}, \tag{34}$$

so $g_n$ converges uniformly to $g$. $\square$

**Lemma 3.** *There exists a homeomorphism*

$$\psi : \mathcal{P}(\Omega) \rightarrow \mathcal{P}_{lin}(\Omega) \tag{35}$$

*where $\mathcal{P}(\Omega)$ is equipped with the weak-* topology with respect to $C(\Omega)$ and $\mathcal{P}_{lin}(\Omega)$ is equipped with the weak-* topology with respect to $C_{lin}(\Omega)$.*

*Proof.* We define $\psi$ and its inverse by

$$\psi(\mu) := \mu - \mu(*)\delta_* \quad \text{and} \quad \psi^{-1}(\nu) = \nu + (1 - \nu(\Omega))\delta_*, \tag{36}$$

where $\delta_*$ denotes the Dirac measure on $* \in \Omega$. By definition of $\mathcal{P}_{\text{lin}}(\Omega)$ in Equation (32), this map is a bijection. It remains to show that $\mu_n \rightarrow \mu$ in $\mathcal{P}(\Omega)$ if and only if $\psi(\mu_n) \rightarrow \psi(\mu)$ in $\mathcal{P}_{\text{lin}}(\Omega)$.

Note that for any $f \in C(\Omega)$ and $\mu \in \mathcal{P}(\Omega)$, we can decompose the integral as

$$\mu(f) = \int_\Omega f d\mu = \int_{\Omega - \{*\}} f d\mu + f(*)\mu(\{*\}). \tag{37}$$

Then, for $f \in C_{\text{lin}}(\Omega)$, we have

$$\mu(f) = \int_{\Omega - \{*\}} f d\mu = \int_{\Omega - \{*\}} f d\psi(\mu) \tag{38}$$

since $f(*) = 0$ and $\mu = \psi(\mu)$ on $\Omega - \{*\}$. Thus, if $\mu_n \rightarrow \mu$ in $\mathcal{P}(\Omega)$ then $\psi(\mu_n) \rightarrow \psi(\mu)$ in $\mathcal{P}_{\text{lin}}(\Omega)$.

Next, suppose that $\nu_n \rightarrow \nu$ in $\mathcal{P}_{\text{lin}}(\Omega)$. Now, we note that $\mathcal{P}_{\text{lin}}(\Omega) \subset \mathcal{M}(\Omega)$, and thus are also continuous functionals on $C(\Omega)$ with respect to the uniform topology. By Lemma 2, we have $\nu_n(f) \rightarrow \nu(f)$ for all $f \in C_0(\Omega)$. However, this also implies it holds for all $f \in C(\Omega)$, since

we obtain functions in $C(\Omega)$ by adding a constant to a function in $C_0(\Omega)$ and $\nu(\{*\}) = 0$ for all $\nu \in \mathcal{P}_{\text{lin}}(\Omega)$. This implies that

$$\int_{\Omega - \{*\}} f d\psi^{-1}(\nu_n) \to \int_{\Omega - \{*\}} f d\psi^{-1}(\nu) \tag{39}$$

since $\nu = \psi^{-1}(\nu)$ on $\Omega - \{*\}$ for all $\nu \in \mathcal{P}_{\text{lin}}(\Omega)$. Furthermore, this implies that $\nu_n(\Omega) \to \nu(\Omega)$, and thus by definition of $\psi^{-1}$ in Equation (36), we have $\psi^{-1}(\nu_n) \to \psi^{-1}(\nu)$ in $\mathcal{P}(\Omega)$. $\qquad \square$

This result allows us to work wtih $\mathcal{P}(\Omega)$ and $\mathcal{P}_{\text{lin}}(\Omega)$ interchangeably. In particular, note that $\Phi_\Omega(*) = 0 \in \mathcal{H}_\Omega$. This implies that for any $\mu \in \mathcal{P}(\Omega)$, we have

$$\Phi_\Omega(\mu) = \Phi_\Omega(\psi(\mu)). \tag{40}$$

*Proof of Theorem 1.* By Corollary 1, $k_\Omega$ is characteristic with respect to $\mathcal{M}_{\text{lin}}(\Omega)$, and $\mathcal{P}_{\text{lin}}(\Omega) \subset \mathcal{M}_{\text{lin}}(\Omega)$ by Theorem 5. Next, by Lemma 3, we can apply the identity Equation (40) to conclude that $k_\Omega$ is characteristic with respect to $\mathcal{P}(\Omega)$. $\qquad \square$

## C    METRIZING WEAK TOPOLOGY

In this appendix, we provide a detailed discussion of the proof of Theorem 2.

**Maximum Mean Discrepancy (MMD).**    Because $k_\Omega$ is a characteristic kernel on $\Omega$, the *kernel mean embedding*, defined (by an abuse of notation) on $\nu = q_* \mu \in \mathcal{M}_{\text{lin}}(\Omega)$ by

$$\Phi_\Omega : \mathcal{M}_{\text{lin}}(\Omega) \to \mathcal{H}_\Omega, \quad \Phi(\nu) := \mathbb{E}_{z \sim \nu}[\Phi_\Omega(z)], \tag{41}$$

is injective. This induces a metric on $\mathcal{M}_{\text{lin}}(\Omega)$, called the *maximum mean discrepancy (MMD)*,

$$\text{MMD}_k(\nu_1, \nu_2) := \left\| \Phi(\nu_1) - \Phi(\nu_2) \right\|_{\mathcal{H}_\Omega}. \tag{42}$$

Our next result shows that the MMD metrizes the weak-* topology on $\mathcal{P}_{\text{lin}}(\Omega)$, where we can directly apply (Sriperumbudur, 2016, Theorem 3.2). See also (Simon-Gabriel et al., 2023) for related results.

**Theorem 6.** *The MMD metric metrizes the weak topology on $\mathcal{P}(\Omega)$. In other words, given measures $\nu_n, \nu \in \mathcal{P}(\Omega)$, we have $MMD_k(\nu_n, \nu) \to 0$ if and only if*

$$|\nu_n(g) - \nu(g)| \to 0 \tag{43}$$

*for all $g \in C(\Omega)$.*

*Proof.* This follows from (Sriperumbudur, 2016, Theorem 3.2) as $\Omega$ is a compact Polish space and $k_\Omega$ is a continuous bounded kernel. $\qquad \square$

**Corollary 2.** *The $p$-Wasserstein metric $W_p$ and the MMD metric $MMD_k$ induce the same topology on $\mathcal{P}(\Omega)$.*

*Proof.* This is a direct consequence of the above, since the $p$-Wasserstein distance also metrizes the weak topology on $\mathcal{P}(\Omega)$ (Villani, 2009, Theorem 6.9). $\qquad \square$

## D    PPM REGULARIZER HAS CONTINUOUS GRADIENTS

We say that a function $f : \mathbb{R}^n \to \mathbb{R}^m$ is a $C^1$ function if all first derivatives of $f$ are continuous. We will begin with a more general statement which will immediately imply Theorem 3. Fix $\mu \in \mathcal{P}_c(\mathbb{R}^N)$ to be a measure, and let $h_\theta : \mathbb{R}^N \to \mathbb{R}^L$ be a mapping from $\mathbb{R}^N$ to the latent space $\mathbb{R}^L$. Suppose that the mapping $h_\theta$ is parametrized by $\theta \in \mathbb{R}^P$, and define $H : \mathbb{R}^P \times \mathbb{R}^N \to \mathbb{R}^L$ by $H(\theta, x) = h_\theta(x)$. We aim to show that $\mathfrak{H} : \mathbb{R}^P \to \mathcal{H}_\Omega$, defined by

$$\mathfrak{H}(\theta) := \Phi_\Omega(\text{PPM}_q(h_\theta(\mu))) \tag{44}$$

is a smooth function. In particular, the pipeline for computing this feature map is

$$\mathcal{P}_c(\mathbb{R}^N) \xrightarrow{(h_\theta)_*} \mathcal{P}_c(\mathbb{R}^L) \xrightarrow{(-)^{\otimes(2k+2)}} \mathcal{P}_c((\mathbb{R}^L)^{2k+2}) \xrightarrow{(\mathrm{PH}_k)_*} \mathcal{P}(\Omega) \xrightarrow{\Phi} \mathcal{H}, \tag{45}$$

Let $F : \mathbb{R}^P \to \mathcal{P}((\mathbb{R}^L)^{2k+2})$ be the map from the parameter space to the product measure in latent space. We assume that the resulting product measure has a $C^1$ density, which is true if $H$ is $C^1$, and $\mu$ has a $C^1$ density. Then, $F$ has the form

$$F(\theta) = f(\theta, x)dx \tag{46}$$

where $f : \mathbb{R}^p \times (\mathbb{R}^L)^{2k+2} \to \mathbb{R}$ is a $C^r$ function.

**Theorem 7.** *Let $\mu \in \mathcal{P}(\mathbb{R}^N)$ be a probability measure with a $C^1$ density, suppose $H : \mathbb{R}^P \times \mathbb{R}^N \to \mathbb{R}^L$ is a $C^1$ function. Then $\mathfrak{H} : \mathbb{R}^P \to \mathcal{H}_\Omega$ is a $C^1$ function (where derivatives are Fréchet derivatives).*

*Proof.* Expanding out the definition of $\mathfrak{H}$, we have

$$\Phi_\Omega(\mathrm{PPM}_q(h_\theta(\mu))) = \int_\Omega \Phi_\Omega(z)d(\mathrm{PH}_k)_*F(\theta)(z) \tag{47}$$

$$= \int_{(\mathbb{R}^L)^{2k+2}} \Phi_\Omega \circ \mathrm{PH}_k(x)dF(\theta)(x) \tag{48}$$

$$= \int_{(\mathbb{R}^L)^{2k+2}} \Phi_\Omega \circ \mathrm{PH}_k(x)f(\theta, x)dx, \tag{49}$$

where we use the definition of the pushforward in the second line, and the definition of the density of $F$ in Equation (46). Now, this integral is a Bochner integral since it is valued in a Hilbert space, and we can still differentiate under the integral (by Hille's theorem; see (Dieudonne, 1969, Paragraph 8.11.2)). Then, we have

$$\frac{\partial}{\partial \theta_i}\Phi_\Omega(\mathrm{PPM}_q(h_\theta(\mu))) = \int_{(\mathbb{R}^L)^{2k+2}} \Phi \circ \mathrm{PH}_k(x)\frac{\partial f(\theta, x)}{\partial \theta_i}dx, \tag{50}$$

which is continuous since $f$ is $C^1$. $\qquad\square$

*Proof of Theorem 3.* By direct application of Theorem 7, both

$$\boldsymbol{\theta} \mapsto \Phi_\Omega \circ \mathrm{PPM}_q \circ d_{\boldsymbol{\theta}}(\mu) \quad \text{and} \quad (\boldsymbol{\theta}, \boldsymbol{\omega}) \mapsto \Phi_\Omega \circ \mathrm{PPM}_q \circ d_{\boldsymbol{\theta}} \circ g_{\boldsymbol{\omega}}(\nu) \tag{51}$$

are $C^1$ functions. Then, since the norm is continuously differentiable away from the origin, we obtain the desired result. $\qquad\square$

# E  SHAPE MATCHING

## E.1  IMPLEMENTATION DETAILS FOR SHAPE MATCHING

**Shape Matching Experiment.** The task is to match the "shape" of two point clouds by optimizing a loss function of the form $\mathcal{L} + \mathcal{T}$ in ambient space. Throughout the experiment, we use gradient descent with momentum as the optimization algorithm. The value of the momentum parameter is 0.9 and the step size is 0.05. Two simple reference shapes in $\mathbb{R}^2$ are chosen for visualization purposes. They are a unit circle (left), and the union of two intersecting unit circles (right), shown in Figure 2. A noisy point cloud with 512 points is first initialized with a normally distributed at the origin with a standard deviation of 0.3, we call training shape. The reference shapes are also sampled as a point cloud wiht 512 points. The reference shape is fixed and the training shape changes towards the reference, guided by some loss function. We test for the effectiveness of four loss function, they are Cramer, MMD, Cramer + PPM-Reg and MMD + PPM-Reg. The MMD metric using an RBF kernel (with width $\sigma = 0.1$). Throughout the experiment, the hyperparameter of PPM-Reg is fixed as $\lambda = 1$, $\lambda_0 = 1$, $\lambda_1 = 6000$, $\sigma = 0.1$ and $s = 2000$. For Cramer + PPM-Reg, the weight of the cramer loss is 1.6. For MMD + PPM-Reg, the weight of the MMD loss is 5. In Figure 4, we show the the shapes after 16000 training steps. Even at this stage, we observe that there are several "leftover" points. This is partially due to the choice of the underlying loss functions, and we observe that in (b), (d) and (h), the trained shapes using PPM-Reg largely capture the topological features of the reference shape.

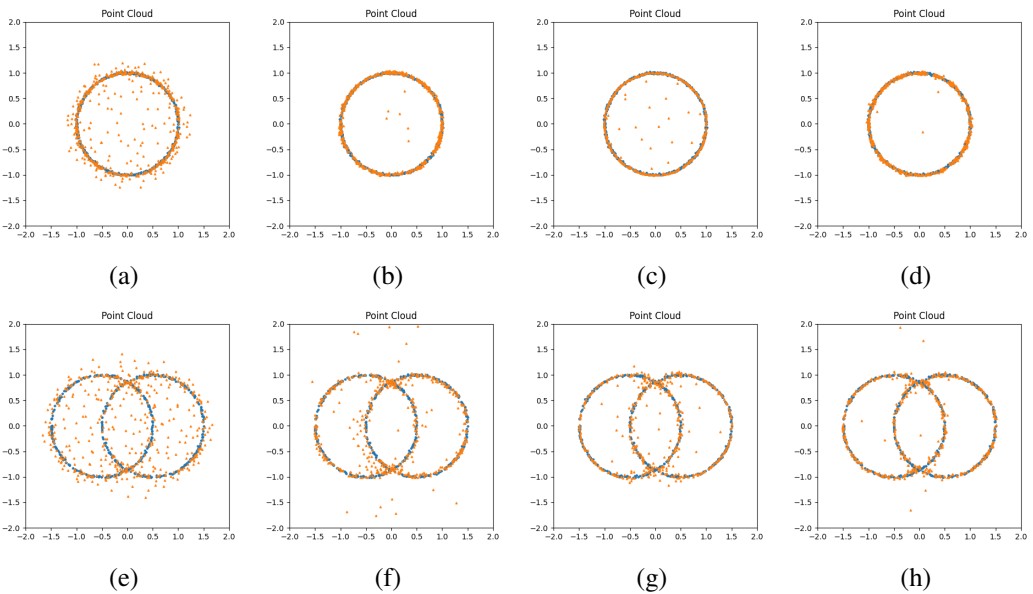

Figure 4: Plots of the shape matching experiment at convergence after 16000 steps. (a) and (e) use only the Cramer loss. (b) and (f) use Cramer + PPM-Reg. (c) and (g) use only the MMD loss. (d) and (h) use MMD + PPM-Reg.

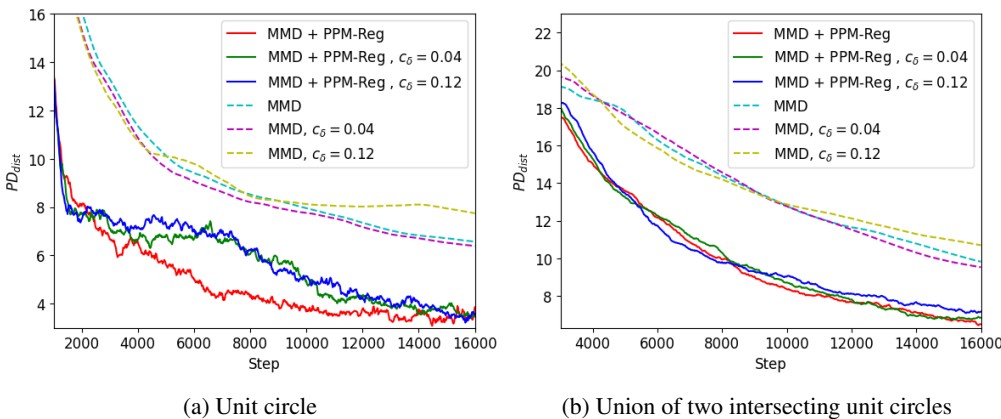

(a) Unit circle

(b) Union of two intersecting unit circles

Figure 5: Illustrative example of the PPM-Reg in shape matching using MMD with increasing handicap contain. Plotting 2-Wasserstein distance upto 1-dimensional persistent homology between the reference shape and training shape ($PD_{dist}$) over optimization steps. The reference shape of (a) unit circle and (b) union of two intersecting unit circles. $c_\delta$ indicate the strength of the handicap (detail provided in Appendix E.2). Showing that the ability to matching topological features as the handicap contains increase.

**Computational Comparison.** Results are computed on Nvidia Geforce RTX 3060 with Intel Core i7-10700.

## E.2   IMPERFECT CONVERGENCE

This section demonstrates the efficacy of PPM-Reg when the primary loss function $\mathcal{L}$ is optimized imperfectly. As discussed in Section 1, most GAN training algorithms converge to a local saddle point. In simpler latent-space matching tasks, $\mathcal{L}$ is often optimized alongside with other tasks. To summarize, converging to an imperfect $\mathcal{L}$ value is a common occurrence in machine learning.

To restrict the solution of the shape matching problem, we impose a penalty term that prevents the centroid of the reference shapes and the training shape from being smaller than a user-defined value $c_\delta$. The penalty term $f_p$ is given by

$$f_p = \frac{\lambda_p}{\beta} ln(1 + e^{\beta(c_\delta - \|c_{train} - c_{ref}\|_2)}), \tag{52}$$

where $c_{train}$ and $c_{ref}$ refers to the centroid of the training shape and reference shapes respectively. The term $\lambda_p$ is the penalty strength, $\beta$ is a tuning parameter. For the MMD case, $\mathcal{V} = \mathcal{L} + f_p$. For the MMD + PPM-Reg case, $\mathcal{V} = \mathcal{L} + \lambda\mathcal{T} + f_p$.

**Implementation Details.** We largely follow the setup on Section 6.2 and Appendix E.1, but make a few minor changes. We reduced the step size to 0.01 and performed 16,000 gradient steps. The hyperparameter of PPM-Reg is fixed as $\lambda = 1$, $\lambda_0 = 0.3$, $\lambda_1 = 6000$, $\sigma = 0.1$ and $s = 2000$. For the penalty function $f_p$, we set $\beta = 80$. The $\lambda_p$ value remains the same when we vary $c_\delta$ and compare against adding PPM-Reg. The $\lambda_p$ value is determined such that $\|c_{train} - c_{ref}\|_2$ converges to $c_\delta$ when $c_\delta = 0.04$.

**Evaluation Metrics.** We consider the case where $c_\delta \in \{0, 0.04, 0.12\}$. For reference shape normalized to $[-1, 1]$, having a $c_\delta = 0.12$ is very small. We compare the main loss with the addition of PPM-Reg and track the 2-Wasserstein distance up-to the 1-dimensional persistence diagrams along gradient step.

**Result.** Figure 5 shows the change in PD distance as $c_\delta$ increases with the circle Figure 5(a) and union of two circles Figure 5(b). Compared with only using MMD as main loss, adding PPM-Reg consistently converges to a smaller PD distance regardless of the $c_\delta$ value. In contrast, when only using MMD, the PD distance increases as $c_\delta$ value increases. Figure 5 illustrates the benefit of *explicitly* comparing the difference between topological features with PPM-Reg compared with *implicitly* considering topological features with only MMD when the optimization problem may not converge to near zero.

# F UNCONDITIONAL IMAGE GENERATION

## F.1 IMPLEMENTATION DETAILS FOR UNCONDITIONAL IMAGE GENERATION

This section fills in the details of the unconditional image generation experiment in Section 6.3.

| | $\sigma = 0.05$ | | | $\sigma = 0.5$ | | | $\sigma = 1.0$ | | |
|---|---|---|---|---|---|---|---|---|---|
| $\lambda$ | CMMD | FD$_{Dinov2}$ | WD$_{latent}$ | CMMD | FD$_{Dinov2}$ | WD$_{latent}$ | CMMD | FD$_{Dinov2}$ | WD$_{latent}$ |
| 1.0 | 0.74 | 945.27 | 0.6330 | 0.68 | 846.33 | 0.6228 | 0.56 | 780.68 | 0.6080 |
| 5.0 | 0.73 | 928.60 | 0.6310 | 0.72 | 884.40 | 0.6282 | 0.74 | 894.77 | 0.6308 |
| 10.0 | 0.74 | 880.68 | 0.6332 | 0.77 | 922.58 | 0.6379 | 0.71 | 923.50 | 0.6274 |

Table 4: Ablation study on AnimeFace dataset.

| | $\sigma = 0.05$ | | | $\sigma = 0.5$ | | | $\sigma = 1.0$ | | |
|---|---|---|---|---|---|---|---|---|---|
| $\lambda$ | CMMD | FD$_{Dinov2}$ | WD$_{latent}$ | CMMD | FD$_{Dinov2}$ | WD$_{latent}$ | CMMD | FD$_{Dinov2}$ | WD$_{latent}$ |
| 1.0 | 0.87 | 737.09 | 0.6945 | 0.65 | 704.74 | 0.6744 | 0.81 | 745.39 | 0.6886 |
| 5.0 | 0.72 | 733.66 | 0.6815 | 0.58 | 700.73 | 0.6666 | 0.68 | 695.36 | 0.6768 |
| 10.0 | 0.61 | 690.01 | 0.6691 | 0.69 | 683.35 | 0.6781 | 0.71 | 719.39 | 0.6795 |

Table 5: Ablation study on CelebA dataset.

**Implementation Details.** The network architectures used in the unconditional image generation in Section 6.3 are shown in Figure 8. As illustrated in Figure 8, $g_{\boldsymbol{\omega}}$ is a ResNet based CNN that takes 128-dimensional noise vector as input. $d_{\boldsymbol{\theta}}$ is a CNN that outputs a 128-dimensional latent vector.

We compare the basic Cramer (Bellemare et al., 2017) value function $\mathcal{V} = \mathcal{L}$, with the use of PPM-Reg $\mathcal{V} = \mathcal{L} + \mathcal{T}$. For the addition of PPM-Reg case, we fix $\lambda_0 = 0.001$, $\lambda_1 = 0.6$, and $s = 1024$. $\lambda = \{1.0, 5.0, 10.0\}$ and $\sigma = \{0.05, 0.1, 0.5\}$ are the tuning parameter. In both cases, the standard Adam optimizer with learning rate $1 \times 10^{-4}$ is used to train the network. For $g_{\boldsymbol{\omega}}$, $\beta_1 = 0.0$ and $\beta_2 = 99$. For $d_{\boldsymbol{\theta}}$, $\beta_1 = 0.5$ and $\beta_2 = 0.99$. The batch size is 192. For the CelebA dataset, the GAN training run for 5440 epochs. For AnimeFace data set, the GAN training run for 7000 epochs.

During training, we compute CMMD for every 160 epochs. We report the CMMD (Jayasumana et al., 2024) and $WD_{\text{latent}}$ and $FD_{\text{Dinov2}}$ (Stein et al., 2024) with the smallest CMMD value across training epochs. Those quantitative metrics are computed by sampling 10K images from the data set and generating 10k images from the network.

**Ablation Study.** There are two primary parameters in our topological regularizer. The parameter $\lambda$ controls the strength of the regularizer, while $\sigma$ controls the width of the RBF kernel in defining the MMD for PPMs. We provide an ablation study to show how the evaluation metrics change as we vary these parameters in Table 5 for AnimeFace and Table 4 for CelebA.

## F.2 FURTHER UNCONDITIONAL IMAGE GENERATION EXPERIMENT

This section introduces supplementary unconditional image generation experiments with an increase in image resolution as well as additional datasets in conjunction with appropriate network architectures.

| | CelebA | | | LSUN Kitchen | | |
|---|---|---|---|---|---|---|
| | CMMD | $FD_{\text{Dinov2}}$ | $WD_{\text{latent}}$ | CMMD | $FD_{\text{Dinov2}}$ | $WD_{\text{latent}}$ |
| Cramer (Bellemare et al., 2017) | 0.52 | 902.09 | 0.7335 | 1.56 | 1592.06 | 0.7690 |
| Cramer + PPM-Reg | 0.46 | 826.04 | 0.7296 | 1.31 | 1381.22 | 0.7502 |

Table 6: Quantitative evaluation on $64 \times 64$ image generation, values are reported at the epoch with the smallest CMMD.

**Implementation Details.** Similar to Section 6.3, $g_{\boldsymbol{\omega}}$ is a ResNet based CNN that takes 128-dimensional noise vector as input. $d_{\boldsymbol{\theta}}$ is a CNN that outputs a 128-dimensional latent vector. The major difference is instead of just generating $32 \times 32$ images as in Section 6.3, we test our PPM-Reg with higher resolution. Specifically, we consider the $64 \times 64$ image generation task. To accommodate the increase in modeling complexity, we introduce a new network architecture shown in Figure 9.

To avoid confusion, we will write out our implementation details. We are comparing Cramer (Bellemare et al., 2017) and the addition of PPM-Reg. For the addition of PPM-Reg case, we only fix $\lambda_0 = 0.001$ and $s = 1024$. The value of $\lambda_1$ changes as the training epoch runs. Specifically, we adapt cosine annealing (Loshchilov & Hutter, 2016) on $\lambda_1$, the $\lambda_1$ value at epoch $t$ term $\lambda_1^t$ is given by

$$\lambda_1^t = \begin{cases} \lambda_1^{min} + \frac{1}{2}(\lambda_1^{max} - \lambda_1^{min})(1 + \cos(\frac{t}{t_{end}}\pi)), & \text{if } t \leq t_{end} \\ \lambda_1^{min}, & \text{if } t > t_{end}, \end{cases} \quad (53)$$

where $\lambda_1^{min}$, $\lambda_1^{max}$ and $t_{end}$ are user-defined variable. $\lambda_1^{min}$ and $\lambda_1^{max}$ are the range of the $\lambda_1$, $t_{end}$ is the ending epoch for the cosine annealing. We fix $\lambda_1^{min} = 0.1$, $\sigma = 0.5$ for both dataset. For CelebA, $\lambda = 1$, $\lambda_1^{max} = 1$ and $t_{end} = 1920$. For LSUN Kitchen, $\lambda = 10$, $\lambda_1^{max} = 0.8$, and $t_{end} = 260$. The standard Adam optimizer with learning rate $1 \times 10^{-4}$ is used to train the network. For $g_{\boldsymbol{\omega}}$, $\beta_1 = 0.0$ and $\beta_2 = 99$. For $d_{\boldsymbol{\theta}}$, $\beta_1 = 0.5$ and $\beta_2 = 0.99$. The batch size is 192. For the CelebA dataset, the GAN training is run for 2560 epochs, while for the LSUN Kitchen data set, the GAN training is run for 300 epochs.

**Dataset and Evaluation Metrics.** We add a new dataset LSUN Kitchen (Yu et al., 2015) and also use CelebA (Liu et al., 2015) at a higher resolution. Images are centered and resized to $64 \times 64$.

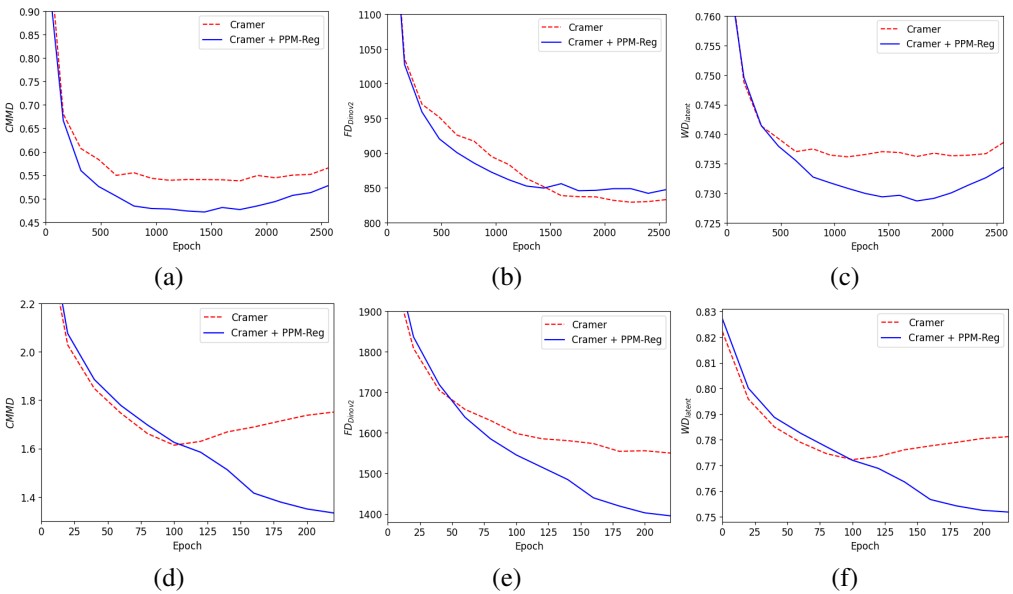

Figure 6: CMMD (a,e), $FD_{Dinov2}$ (b,f) and $WD_{latent}$ (c,g) versus training epochs for the CelebA (a-c) and LSUN Kitchen (d-f) dataset for the $64 \times 64$ image generation. 10K samples are randomly generated to compute the distances, and moving averages with a window of 5 are used to smooth the values. For CelebA, distances are recorded every 160 epochs. For LSUN Kitchen, distances are recorded every 20 epochs.

During training, we compute CMMD for every 160 epochs for CelebA and 20 epochs for LSUN Kitchen. We report the CMMD (Jayasumana et al., 2024) and $WD_{latent}$ and $FD_{Dinov2}$ (Stein et al., 2024) with the smallest CMMD value across training epochs. For justification for using the chosen quantitative evaluation metrics, readers can refer to Section 6.3. Those quantitative metrics are computed by sampling 10K images from the data set and generating 10k images from the network.

**Result.** Figure 6 tracks the three quantitative metrics during training and Table 6 reports the three metrics with the smallest CMMD. While the $FD_{Dinov2}$ metric has comparable values in Figure 6(b), at the point of the smallest CMMD, Cramer + PPM-Reg has a smaller $FD_{Dinov2}$ for CelebA, shown in Table 6. Our quantitative results in Table 6 reinforce the fact that using PPM-Reg improves the generative ability of GANs. Compared to CelebA dataset, using PPM-Reg in the LSUN Kitchen dataset results in a more significant improvement. We conjecture that since LSUN Kitchen is a much larger dataset (2,212,277 training samples) that contains diverse images (i.e. different color tones, layouts), its underlying lower dimensional submanifold has more complex topological features compared with CelebA. The significant improvement gives evidence that PPM-Reg is useful in discovering more complex topological structures in latent space.

## G  SEMI-SUPERVISED LEARNING

### G.1  IMPLEMENTATION DETAILS FOR SEMI-SUPERVISED LEARNING

This section fills in the details of the semi-supervised learning experiment in Section 6.4.

**Network Architecture and Implementation Details.**    The network architectures used in the semi-supervised learning experiment in Section 6.4 are shown in Figure 10. The input noise vector of $g_{\omega}$ has dimension of 64. We set the dimension of the output latent vector of $d_{\theta}$ as 64.

Specifically, $g_{\omega}$ and $d_{\theta}$ are trained with the GAN framework for 4,000 epochs using the standard Adam optimizer with learning rate $1 \times 10^{-4}$. For $g_{\omega}$, we set $\beta_1 = 0.0$ and $\beta_2 = 99$; for $d_{\theta}$, we set $\beta_1 = 0.5$ and $\beta_2 = 0.99$. The batch size is 192. For Cramer + PPM-Reg, we fix $\lambda_0 = 1$, $\lambda_1 = 90$

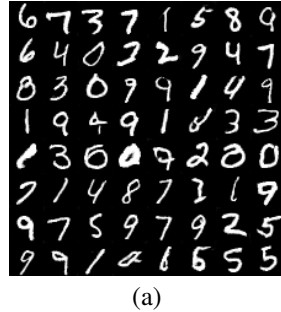 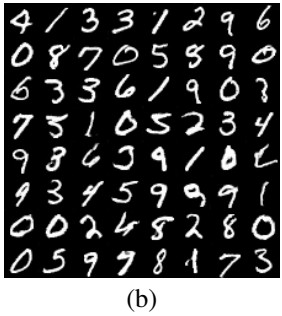

(a)                              (b)

Figure 7: Generated images from SSL experiment from trained $g_{\boldsymbol{\omega}}$ for MNIST using (a) Cramer Distance and (b) PPM-Reg.

and $s = 1024$. $\lambda = \{0.025, 0.05, 0.1\}$ and $\sigma = \{0.05, 0.1, 0.5\}$ are the tuning parameter. After the GAN framework completes the training of $g_{\boldsymbol{\omega}}$ and $d_{\boldsymbol{\theta}}$, the weights of $d_{\boldsymbol{\theta}}$ are frozen. Using $d_{\boldsymbol{\theta}}$ as feature extraction, the output of $d_{\boldsymbol{\theta}}$ is fed into the classifier. The classifier $c_{\gamma}$ is train with the standard Adam optimizer with learning rate $1 \times 10^{-4}$ where $\beta_1 = 0.1$ and $\beta_2 = 0.99$ for 1000 epochs.

"Baseline" connects $d_{\boldsymbol{\theta}}$ and $c_{\gamma}$ and trains as a classifier without first pertaining $d_{\boldsymbol{\theta}}$ with GANs. The standard Adam optimizer with learning rate $1 \times 10^{-4}$ where $\beta_1 = 0.1$ and $\beta_2 = 0.99$ are use to train the "Baseline" network for 4,000 epochs.

|  | Fashion-MNIST (200 labels) | | | Fashion-MNIST (400 labels) | | |
|---|---|---|---|---|---|---|
| $\lambda$ | $\sigma = 0.05$ | $\sigma = 0.1$ | $\sigma = 0.5$ | $\sigma = 0.05$ | $\sigma = 0.1$ | $\sigma = 0.5$ |
| 0.1 | $75.39 \pm 1.12$ | $74.96 \pm 0.96$ | $75.24 \pm 1.20$ | $75.39 \pm 1.12$ | $74.96 \pm 0.96$ | $75.24 \pm 1.20$ |
| 0.05 | $76.35 \pm 1.09$ | $76.25 \pm 1.30$ | $76.81 \pm 0.88$ | $76.35 \pm 1.09$ | $76.25 \pm 1.30$ | $76.81 \pm 0.88$ |
| 0.025 | $76.42 \pm 0.85$ | $76.77 \pm 1.29$ | $76.84 \pm 1.23$ | $76.42 \pm 0.85$ | $76.77 \pm 1.29$ | $76.84 \pm 1.23$ |

Table 7: Ablation study on Fashion-MNIST dataset with 200/400 labels. Test-set classification accuracy (%) is shown averaged over 10 runs.

|  | Kuzushiji-MNIST (200 labels) | | | Kuzushiji-MNIST (400 labels) | | |
|---|---|---|---|---|---|---|
| $\lambda$ | $\sigma = 0.05$ | $\sigma = 0.1$ | $\sigma = 0.5$ | $\sigma = 0.05$ | $\sigma = 0.1$ | $\sigma = 0.5$ |
| 0.1 | $73.70 \pm 2.58$ | $70.84 \pm 2.56$ | $70.16 \pm 1.96$ | $78.35 \pm 1.81$ | $76.18 \pm 1.87$ | $76.67 \pm 1.31$ |
| 0.05 | $74.35 \pm 2.45$ | $73.65 \pm 1.70$ | $73.97 \pm 1.24$ | $78.99 \pm 1.41$ | $79.92 \pm 1.88$ | $78.77 \pm 1.48$ |
| 0.025 | $74.47 \pm 1.65$ | $75.78 \pm 1.99$ | $75.41 \pm 2.83$ | $79.40 \pm 1.65$ | $79.33 \pm 1.69$ | $80.04 \pm 1.37$ |

Table 8: Ablation study on Kuzushiji-MNIST with 200/400 labels. Test-set classification accuracy (%) is shown averaged over 10 runs.

**Ablation Study.** We show how the classification accuracy varies with respect to the topological regularization strength $\lambda$ and the RBF width $\sigma$ in Table 7 for Fashion-MNIST, Table 8 for Kuzushiji-MNIST, and Table 9 for MNIST.

### G.2 Further Semi-Supervised Learning Experiment

This section introduces supplementary semi-supervised learning experiments with additional datasets in conjunction with appropriate network architectures.

**Network Architecture and Implementation Details.** Similar to the setup in Section 6.4, $g_{\boldsymbol{\omega}}$ is a deconvolutional network with 64 dimension noise vector as input. $d_{\boldsymbol{\theta}}$ is a CNN with a 64 dimension latent vector as output. The additional SVHM dataset is more intricate than the MNIST variants evaluated in Section 6.4, as it is a color image dataset with higher resolution. To tackle the increase in

| $\lambda$ | MNIST (200 labels) | | | MNIST (400 labels) | | |
|---|---|---|---|---|---|---|
| | $\sigma = 0.05$ | $\sigma = 0.1$ | $\sigma = 0.5$ | $\sigma = 0.05$ | $\sigma = 0.1$ | $\sigma = 0.5$ |
| 0.1 | $96.34 \pm 0.29$ | $96.32 \pm 0.20$ | $96.18 \pm 0.19$ | $97.19 \pm 0.28$ | $97.06 \pm 0.28$ | $97.07 \pm 0.26$ |
| 0.05 | $96.61 \pm 0.37$ | $96.62 \pm 0.39$ | $96.22 \pm 0.59$ | $97.29 \pm 0.20$ | $97.33 \pm 0.21$ | $97.16 \pm 0.25$ |
| 0.025 | $96.13 \pm 0.42$ | $95.97 \pm 0.37$ | $95.91 \pm 0.55$ | $97.04 \pm 0.27$ | $96.92 \pm 0.20$ | $97.2 \pm 0.25$ |

Table 9: Ablation study on MNIST with 200/400 labels. Test-set classification accuracy (%) is shown averaged over 10 runs.

input dimension and the complexity of modeling the dataset, we introduce a new network architecture shown in Figure 11.

The training setup largely follows the previous experiment; the major difference is the number of training epochs. Specifically, $g_{\boldsymbol{\omega}}$ and $d_{\boldsymbol{\theta}}$ are trained with the GAN framework for 1,200 epochs. The standard Adam optimizer with learning rate $1 \times 10^{-4}$ is used to train the network. For $g_{\boldsymbol{\omega}}$, $\beta_1 = 0.0$ and $\beta_2 = 99$. For $d_{\boldsymbol{\theta}}$, $\beta_1 = 0.5$ and $\beta_2 = 0.99$. The batch size is 192. For Cramer + PPM-Reg, we fix $\lambda_0 = 1$, $\lambda_1 = 90$ and $s = 1024$. $\lambda = \{0.025, 0.05, 0.1\}$ and $\sigma = \{0.05, 0.1, 0.5\}$ are the tuning parameter. After the GAN framework completes the training of $g_{\boldsymbol{\omega}}$ and $d_{\boldsymbol{\theta}}$, the weights of $d_{\boldsymbol{\theta}}$ are frozen. Using $d_{\boldsymbol{\theta}}$ as feature extraction, the output of $d_{\boldsymbol{\theta}}$ is fed into the classifier. The classifier $c_{\gamma}$ is trained with the standard Adam optimizer with learning rate $1 \times 10^{-4}$ where $\beta_1 = 0.1$ and $\beta_2 = 0.99$ for 1000 epochs. The standard Adam optimizer with learning rate $1 \times 10^{-4}$ where $\beta_1 = 0.1$ and $\beta_2 = 0.99$ are use to train the "Baseline" network for 4,000 epochs.

**Dataset and Evaluation Metrics.** We compare the SSL performance with the dataset SVHN. In this experiment, 400 and 600 labels are randomly sampled from the data set. Because of the random nature involved in selecting a few labels, we conducted the experiments ten times and provided the statistics of the highest test-set accuracy achieved.

| | SVHN | |
|---|---|---|
| Number of labels | 400 | 600 |
| Baseline | $44.68 \pm 2.44$ | $53.68 \pm 4.15$ |
| Cramer | $38.12 \pm 1.75$ | $43.23 \pm 1.14$ |
| Cramer + PPM-Reg | $\mathbf{57.20 \pm 1.51}$ | $\mathbf{61.39 \pm 0.66}$ |

Table 10: Test-set classification accuracy (%) on SVHN with 400 and 600 labeled examples. The average and the error bar are computed over 10 runs.

| $\lambda$ | SVHN (400 labels) | | | SVHN (600 labels) | | |
|---|---|---|---|---|---|---|
| | $\sigma = 0.05$ | $\sigma = 0.1$ | $\sigma = 0.5$ | $\sigma = 0.05$ | $\sigma = 0.1$ | $\sigma = 0.5$ |
| 0.1 | $55.66 \pm 1.78$ | $57.20 \pm 1.51$ | $54.84 \pm 1.21$ | $59.48 \pm 1.26$ | $61.39 \pm 0.66$ | $59.13 \pm 0.95$ |
| 0.05 | $47.56 \pm 1.67$ | $56.24 \pm 1.25$ | $52.41 \pm 1.49$ | $50.73 \pm 1.06$ | $60.08 \pm 0.84$ | $60.91 \pm 1.19$ |
| 0.025 | $55.96 \pm 1.03$ | $56.96 \pm 1.92$ | $55.35 \pm 1.06$ | $59.13 \pm 0.95$ | $56.45 \pm 1.41$ | $58.97 \pm 1.00$ |

Table 11: Ablation study on SVHN with 400/600 labels. Test-set classification accuracy (%) is shown averaged over 10 runs.

**Result.** Table 10 shows the test classification accuracy. An ablation study is provided in Table 11. SVHN contains 72,657 training samples and 400 and 600 labels constitute only 0.55% and 0.82% of the original datasets, respectively. The results follow the same characteristic in Section 6.4, only using Cramer does not improve the classification accuracy in SSL. In contrast, the use of PPM-Reg results in a notable improvement in classification accuracy. Specifically, using PPM-Reg with 400 labels yields a 12.52% increase in accuracy compared to the Baseline. The consistent outcome with more complex datasets and network architecture reinforces our claim that PPM-Reg can help learn a more informative latent encoding thereby improving SSL performance.

# H NETWORK ARCHITECTURE

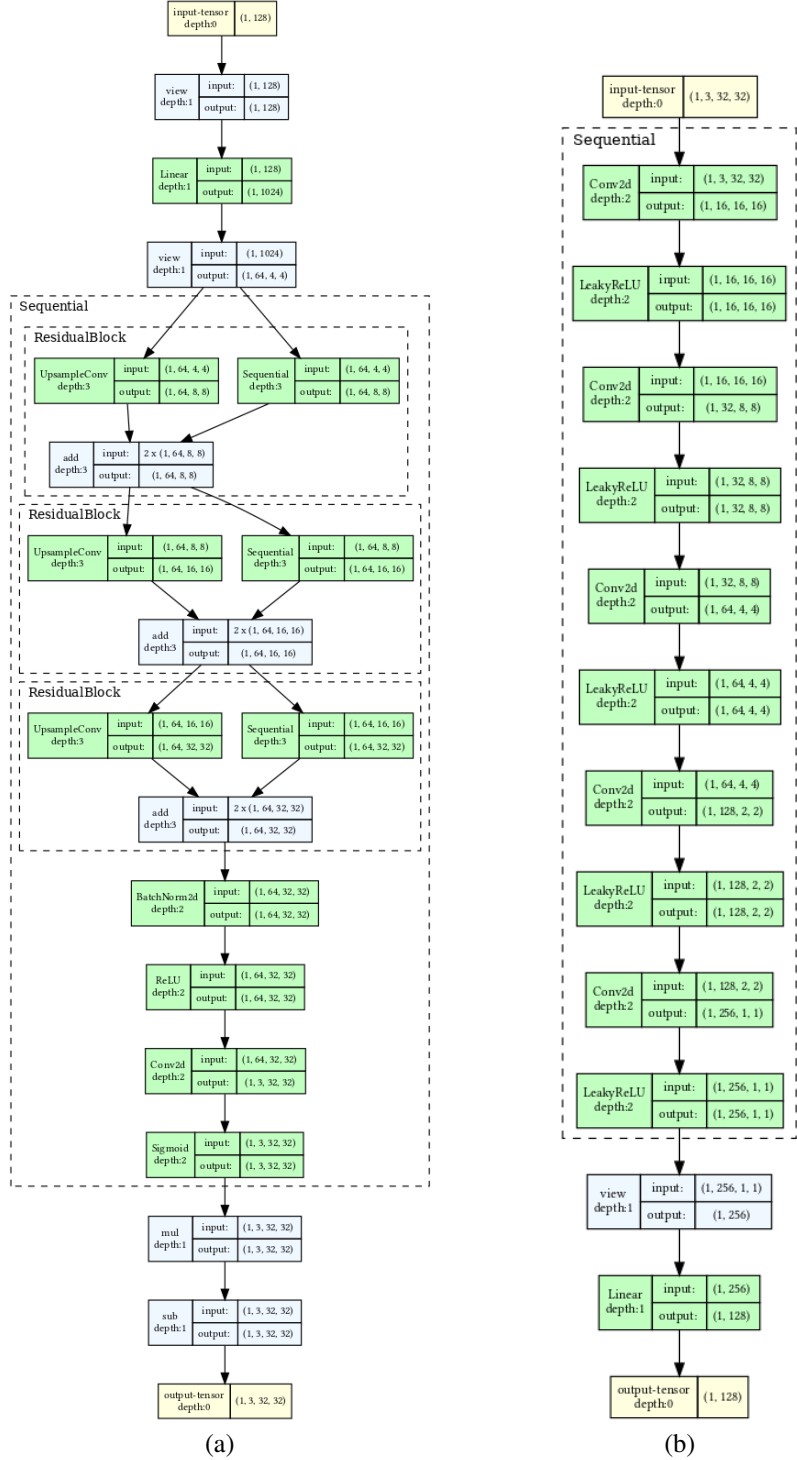

Figure 8: Network architecture of generator (a), discriminator (b) for $32 \times 32$ unconditional image generation.

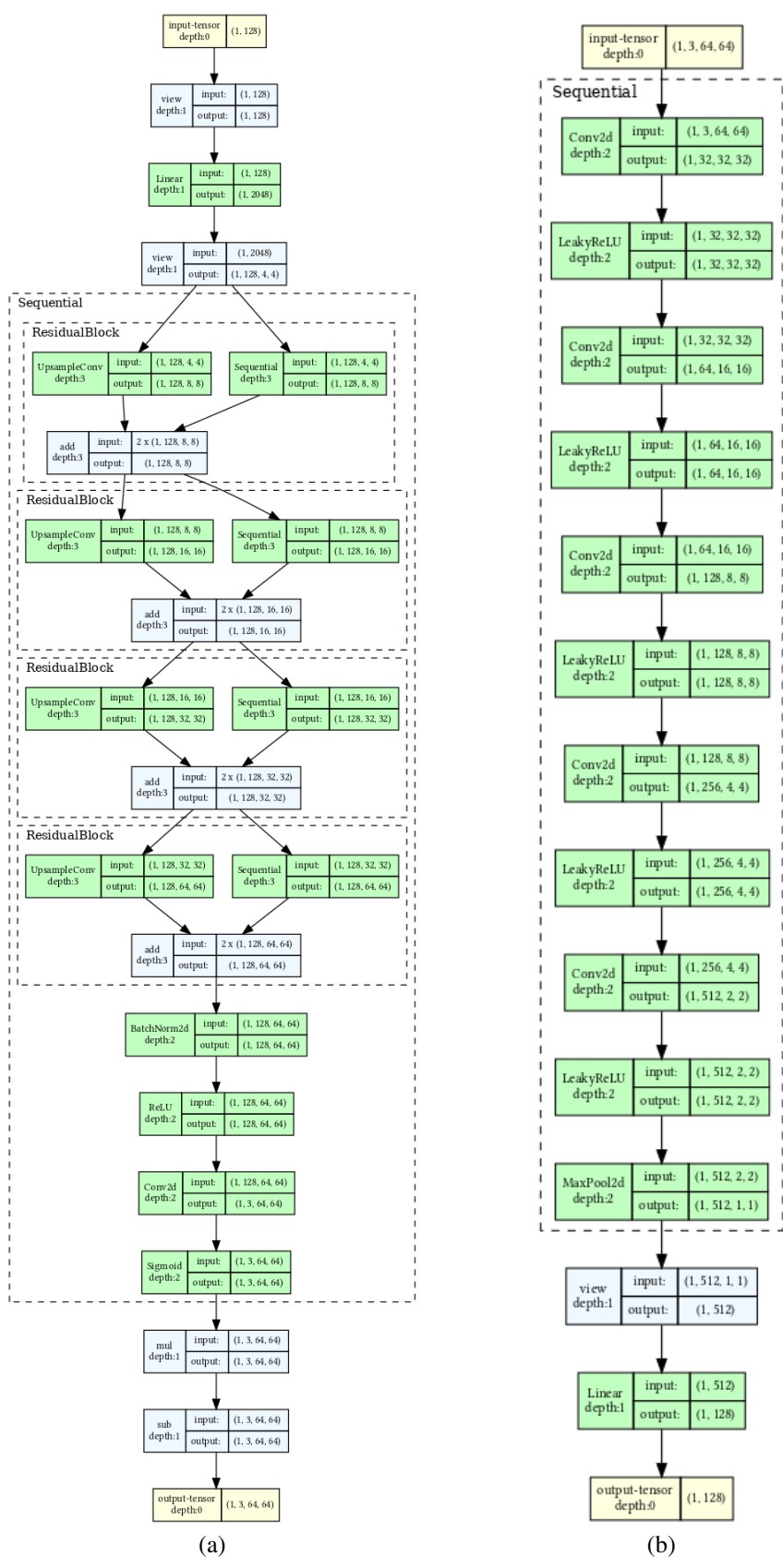

Figure 9: Network architecture of generator (a), discriminator (b) for $64 \times 64$ unconditional image generation.

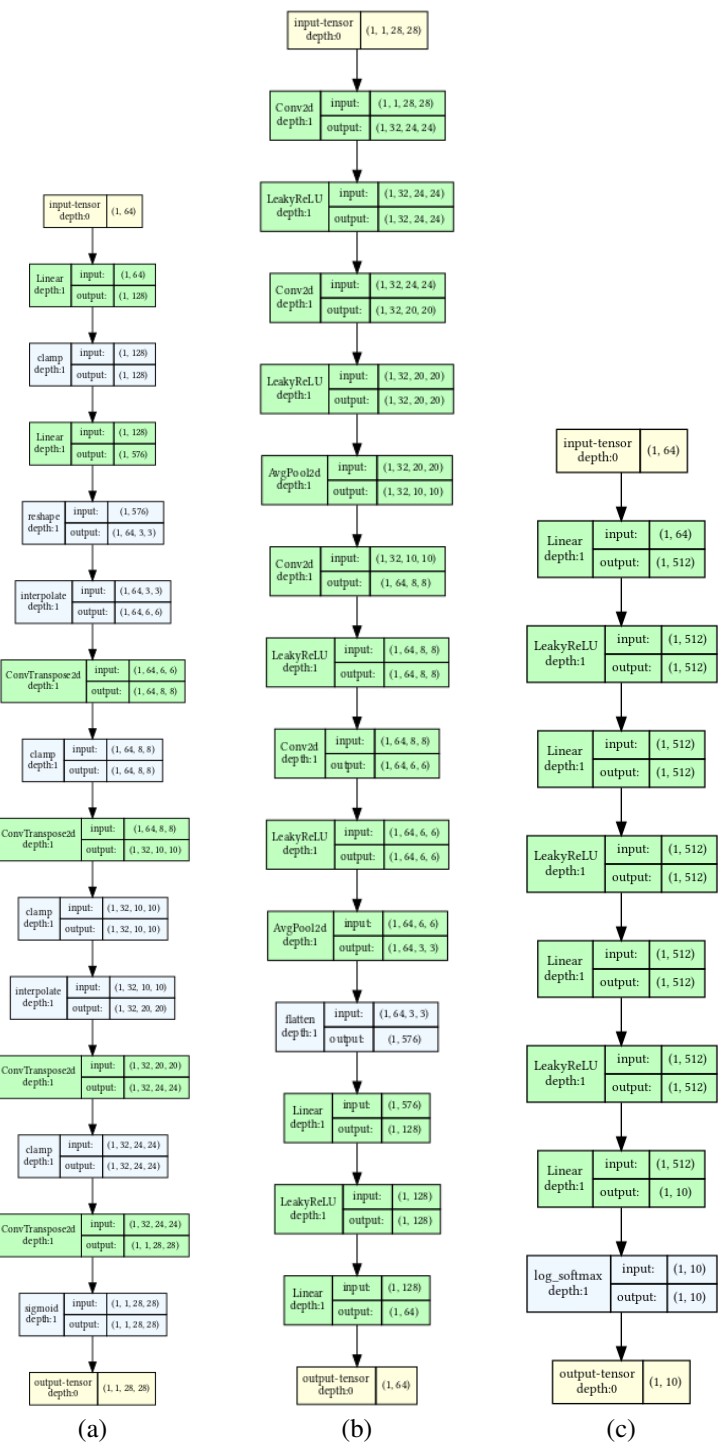

Figure 10: Network architecture of generator (a), discriminator (b) and classifiacter (c) for semi-supervised learning for MNIST variants.

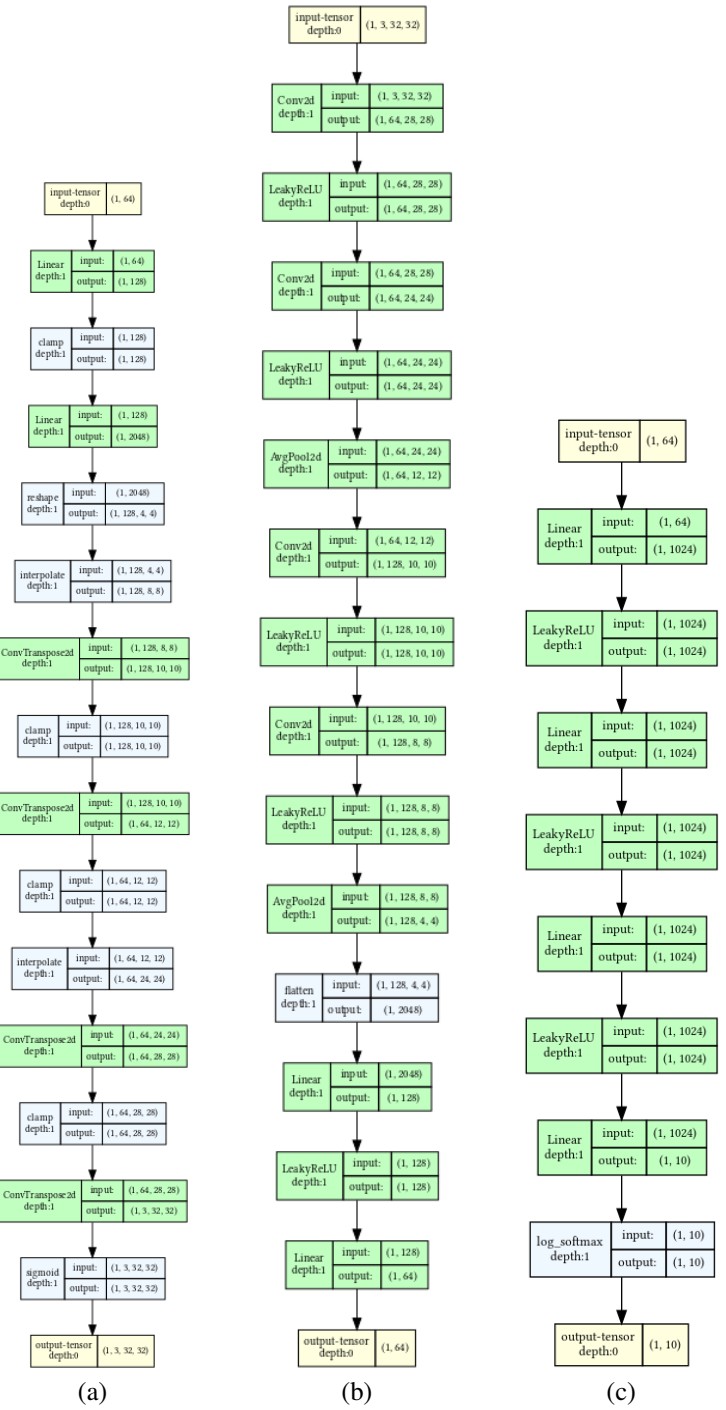

Figure 11: Network architecture of generator (a), discriminator (b) and classifiacter (c) for semi-supervised learning for SVHN dataset.

