# OpenReview forum: "Towards Scalable Topological Regularizers"
_ICLR.cc/2025/Conference — ICLR 2025 Poster_

### Official Review · Reviewer_iUNk · 2024-10-28

**Soundness:** 3
**Presentation:** 3
**Contribution:** 3
**Rating:** 5
**Confidence:** 3

**Summary:**

One of the prohibitive factors of Topological Data Analysis are computational
constraints imposed on the calculation of Persistent Homology (PH). While
highly optimized CPU-GPU algorithms exist, the computation already become
unfeasible for small point clouds in low dimensional settings. A remedy is
proposed through the use of a GPU implementation of the Principal Persistence
Measure (PPM) (Gómez & Mémoli 2024). This addresses both difficulties related
to smoothing as well as scalability. Given the scalable backbone the authors
then provide a theoretical framework to turn the PPM into a regularizer that
can be used in generative tasks where the topology of the dataset is of
importance. They provide a strong theoretical foundation for their regularizer
and show that the gradients are continuous.

The authors provide two experiments as empirical evidence of the efficacy of
their method. First is an optimization task where a point cloud is
backpropagated to match a reference shape. Faster convergence to the final
solution is shown, thereby showing topological regularization to be helpful in
convergence. In a final suite of experiments the authors show that the
regularization significantly increases the generative quality of the trained
models and this is demonstrated in both unsupervised and semi-supervised
learning tasks. The computational advantages are shown to have a significant
increase in training time over the traditional PH calculation and not suffer from the
exponential cost.

**Strengths:**

In general it is great to see a push towards scalable topological methods and
this entails the core strengths of the paper, as it an impactful and relevant
problem to work on. Often in TDA, the focal point lies in theoretical
justification of the methods with only limited emperical evidence, mostly on
small datasets. Lack of scalability of PH being the main driver of this
phenomenon and therefore this paper is of importance in building the bridge in
applying TDA in large scale applications.

The theoretical justification is both thorough and sound and adopts a novel
view on using PH in machine learning applications with the empirical evidence
showing good improvements when using the regularization term.

**Weaknesses:**

Where the theoretical contributions have good exposition, the implementation
details are less thoroughly addressed. As one of the two cornerstones of the
contributions, it would be nice to see where the authors differ in their
implementation from previous work that allowed them to compute the PH on the
GPU. The details seem to be lacking in the paper and together with the release
of the code, would strengthen the paper considerably.

For the experimental section the optimization of point clouds is not
particularly convincing. In the figure the results could be improved by showing
how the final output has converged and in two dimensions, which would be
natural as other metrics are not provided. Especially since the method (I
assume) optimizes only a subset of the whole point cloud at a time I would like
the doubt of only a subset of the point cloud converging to the reference point
cloud to be taken away. Second, only the loss of the training set over the
iterations is shown. An independent set distance to quantify the results would
strengthen the results. The Wasserstein distance between the point clouds or
the Chamfer Distance would be great examples. Finally, an argument why the
non-zero convergence is correct would also be in place, since none of the loss
functions converge to zero.

The experimental section also heavily relies on the Cramer loss, and although
the paper presenting it was rejected at ICLR 2018, although the paper is
well-cited. The reviewer is not an expert on training GAN's and evaluating
them, hence it would be nice to provide an argument as to why this is the main
loss metric to use and also the main model to use and compare to.

If the reviewer would have to show to the reader that a certain regularization
term is effective, a good strategy would be to take a variety of datasets and a
variety of models (perhaps with certain properties) and show that the
regularization consistently improves the scores across the board. As the paper
is currently presented, the scope of the experiments is rather limited.
Moreover, there is also comparable work on topological regularizers for
classification problems which would also provide a nice comparison partner. A
last potential point of improvement is the connection with other topological
methods and regularization terms.
The application of persistent homology is not new and a comparison with previous work,
both in terms of computational performance and increase in accuracy would also be
very nice.

Although the reviewer commends the line of work the authors have decided to
pursue, a considerable amount of work would have to be done to create a logical
and convincing experimental suite. Multiple reference architectures of GAN's
would have to be implemented with more standard metrics. Perhaps other tasks
would also benefit from this regularization term, such as classification and
regression tasks. The non-standard set up for the point cloud optimization
experiment would also have to be convincingly revised. Once these elements are
addressed, the work has great potential impact on the Topological Data Analysis
community and machine learning in general.

**Questions:**

- A question that would also be interesting to consider is how the method
    scales with dimension, based on the paper it looks like it might scale
    exponentially in the dimension.
- What causes loss not to converge to zero in the point cloud optimization
    task?

Remark:
During the review, some typo's were found. Please give the paper another read
and correct them.

---

> ### Comment · Reviewer_iUNk · 2024-11-25
>
> The reviewer would like to thank the authors for their effort in submitting the manuscript. It would also be greatly strengthen the impact of the work if the authors decided to engage with the reviews given.
>
> In my initial review one of the weaknesses regarded the optimization of point clouds. Due to a misread on the reviewers side they only found the animations in the supplementary material yesterday. (It might a good idea to add a before after optimization in the main text, instead of *just* the before to prevent the confusion of the reviewer.)
> Some comments are in place:
> - Answering my own question regarding the loss not being zero at the end is induced by the fact that the point clouds do not converge to their final target in none of the cases. From my own experiments, a different choice of loss term (such as the EMD or CD) would have resulted in much better convergence than either of the two hence raising a question of validity of this  experiment. If needed the reviewer can provide the code for the experiment that proves the claim.
> - It also raises a more fundamental question. If the loss term can not capture the topology of simple objects such as a circle, how will this regularization term capture any type of relevant information in something complex as an image.
> - Also, then point cloud optimization over complex point clouds such as ModelNet40 are out of the question and then the reviewer would like to gain a better understanding of what the regularization term regularizes over.
> - A last point is that the final results do not seem to capture the density of the reference point cloud.
>
> If the authors address the above points it would be great for the discussion. Due to the lack of engagement the score has been reduced and it will be reconsidered if the reviews are adequately addressed.

---

> ### Author Response · Authors · 2024-11-25
> **Initial Comments (1/3)**
>
> We would like to thank the reviewer for taking the time to review our article and provide helpful feedback! Firstly, we apologize for the delay in response. We have been taking all of the reviews into consideration and spending considerable time to address all of the comments. Our original plan was to respond after completing a revision of the paper in order to directly point to how our revision addresses each comment. In particular, we have been performing additional experiments as requested by the reviewer, but this had led to some delays due to limited computational resources. However, we also understand the need for discussion, and hope that our response now can rectify this oversight on our part.
>
> We believe the reviewer may have misunderstood parts of our motivation and the technical content of our article. We apologize for the confusion, and hope that these comments and the forthcoming revision will clarify these issues.
>
> ### Shape Matching Comments
>
> > As one of the two cornerstones of the contributions, it would be nice to see where the authors differ in their implementation from previous work that allowed them to compute the PH on the GPU. The details seem to be lacking in the paper and together with the release of the code, would strengthen the paper considerably.
>
> The main difference is the use of subsampling in principal persistence measures (PPM) rather than computing the full persistence diagram of a point cloud. While computation of the full persistence diagram is computationally expensive, the computation of dimension $q$ persistence on a subsampled point cloud with $2q+2$ points is very simple (Equation 4). In particular, this consists of computing min and max of a small distance matrix -- we compute Equation 4 for $s$ subsamples in parallel on the GPU. We will emphasize this point in our revision. Furthermore, we have included the code in the supplementary material in the original submission, and we will put this in a public github repository if accepted.
>
> >For the experimental section the optimization of point clouds is not particularly convincing. In the figure the results could be improved by showing how the final output has converged and in two dimensions, which would be natural as other metrics are not provided.
>
> Section 6.2 on shape matching of point clouds is meant to be an expository experiment to demonstrate two aspects of our proposed regularizer:
> 1) Our regularizer appropriately takes into account the topological structure of the point clouds.
> 2) Our regularizer is computationally feasible for larger datasets. (Table 1)
>
> We note that the plots in Figure 2 show the Wasserstein distance between the full persistence diagrams (not the PPMs, which is what we are optimizing) -- in particular, we are not showing the training loss. These plots demonstrate that optimizing for PPM effectively accounts for the topological structure of the point clouds.
>
> > Especially since the method (I assume) optimizes only a subset of the whole point cloud at a time I would like the doubt of only a subset of the point cloud converging to the reference point cloud to be taken away.
>
> While we compute persistent homology of subsamples, the PPM computes the persistence of many subsamples in parallel. In this experiment with 512 points, we take 2000 random subsamples of 4 points (see Appendix F), so the method uses the information from all points with high probability.
>
> >Second, only the loss of the training set over the iterations is shown. An independent set distance to quantify the results would strengthen the results.
>
> This is addressed above.
>
> > The Wasserstein distance between the point clouds or the Chamfer Distance would be great examples.
>
> As our goal with this section is to demonstrate that PPM effectively regularizes the *topological* features of the point cloud, we believe comparing the full persistence diagrams is the appropriate metric to show.
>
> > Finally, an argument why the non-zero convergence is correct would also be in place, since none of the loss functions converge to zero.
>
> The reason the final output has not converged is because we are showing the beginning of the optimization. While this is not the focus of this experiment (see comments below), we will include additional figures regarding the convergence in the revised version.
>
> (Continued in next comment)

---

> ### Author Response · Authors · 2024-11-25
> **Initial Comments - Continued (2/3)**
>
> **From the new comment:**
> > In my initial review one of the weaknesses regarded the optimization of point clouds. Due to a misread on the reviewers side they only found the animations in the supplementary material yesterday. (It might a good idea to add a before after optimization in the main text, instead of _just_ the before to prevent the confusion of the reviewer.)
>
> We will edit the text to point out the animations to the reader.
>
> > 1. Answering my own question regarding the loss not being zero at the end is induced by the fact that the point clouds do not converge to their final target in none of the cases. From my own experiments, a different choice of loss term (such as the EMD or CD) would have resulted in much better convergence than either of the two hence raising a question of validity of this experiment. If needed the reviewer can provide the code for the experiment that proves the claim.
> > 2.  It also raises a more fundamental question. If the loss term can not capture the topology of simple objects such as a circle, how will this regularization term capture any type of relevant information in something complex as an image.
>
> As mentioned above, the plots in the initial submission only show the beginning of the optimization step. In particular, we *do* use the Cramer distance as a loss term, but when comparing to the Cramer distance *with* our additional PPM regularization, the topological distance is optimized more quickly (shown in Fig 2). Furthermore, the point of the experiment is not to show that it performs better at convergence. In theory, using a valid metric on the space of probability measures with or without regularization will result in the same limit (if $\mu, \nu$ are probability measures such that $d(\mu, \nu) = 0$, then $\mu = \nu$).
>
> Rather, our goal is to show that adding PPM-Reg will emphasize the topological features during the optimization. This is important in GAN settings where the loss function is changing at every iteration. In our revision, we will include a modified experiment, where the training point cloud has an external constraint which prohibits convergence to the reference point cloud, and demonstrates that topological features are still captured in this setting.
>
> Earth mover distance (EMD) also known as Wasserstein distance is time-consuming to compute as discussed in the introduction. While Chamfer Distance is very popular in point cloud matching, this is not widely used as a loss function in latent space matching as well as GAN training. Because our interest is in popular GAN training frameworks, we mainly demonstrate the use of PPM-Reg with such loss functions in this section.
>
> We plan to revise our exposition of this section to emphasize the points above, and to reduce any confusion on the reader.
>
> (Continued on next comment)

---

> ### Author Response · Authors · 2024-11-25
> **Initial Comment - Continued (3/3)**
>
> ### Other Comments
>
> > The experimental section also heavily relies on the Cramer loss, and although the paper presenting it was rejected at ICLR 2018, although the paper is well-cited. The reviewer is not an expert on training GAN's and evaluating them, hence it would be nice to provide an argument as to why this is the main loss metric to use and also the main model to use and compare to.
>
> Our aim in the experiments is to demonstrate how PPM-Reg can effectively regularize topological features in GAN models. One of the limitations of using Wasserstin distance in GANs is the computational cost. We choose to use the Cramer distance as this is much cheaper to compute, and does not rely on additional hyperparameters to allow for a straightforward comparison. Our experiments show that adding PPM-Reg will improve the performance on such GAN models.
>
>
> > If the reviewer would have to show to the reader that a certain regularization term is effective, a good strategy would be to take a variety of datasets and a variety of models (perhaps with certain properties) and show that the regularization consistently improves the scores across the board. As the paper is currently presented, the scope of the experiments is rather limited. Moreover, there is also comparable work on topological regularizers for classification problems which would also provide a nice comparison partner. A last potential point of improvement is the connection with other topological methods and regularization terms. The application of persistent homology is not new and a comparison with previous work, both in terms of computational performance and increase in accuracy would also be very nice.
> >
> > Although the reviewer commends the line of work the authors have decided to pursue, a considerable amount of work would have to be done to create a logical and convincing experimental suite. Multiple reference architectures of GAN's would have to be implemented with more standard metrics. Perhaps other tasks would also benefit from this regularization term, such as classification and regression tasks.
>
> While we believe that our method can be used in a wider range of experiments, we have chosen to work with GANs to provide a coherent focus to our paper. Our revision will emphasize this in the introduction, and we believe that further experiments in other domains such as classification and regression tasks would be interesting for future work and beyond the scope of the current article.
>
> We have chosen our two experiments as they demonstrate the ability of PPM-Reg to improve the performance on both the *generative* and *discriminative* ability of GANs. We are currently running additional experiments, and will provide more detailed comments when this is complete.
>
> > The non-standard set up for the point cloud optimization experiment would also have to be convincingly revised. Once these elements are addressed, the work has great potential impact on the Topological Data Analysis community and machine learning in general.
>
> We believe our set up for the shape matching experiments is fairly standard in showing the ability of PPM-Reg to take into account the topological features of the point clouds. We acknowledge the reviewer has suggested to show convergence with respect to other metrics such as Cramer, but this would not show the *topological* benefits of our method. We are wondering if there is another component the reviewer believes is non-standard that we could address?
>
> > A question that would also be interesting to consider is how the method scales with dimension, based on the paper it looks like it might scale exponentially in the dimension.
>
> It depends on which dimension the reviewer is referring to. Regarding the **homology dimension** $q$: In this case, each subsample will consist of $2q+2$ points, so the number of entries in the distance matrix will grow quadratically. The expensive step of the computation is Equation 4, where we must compute min and max, which has linear complexity. Overall, this scales quadratically with the homology dimension.
>
> Regarding the **dimension of the point cloud**: This will only affect the computation of the distance matrix, which scales linearly with the dimension. However, this computation is done prior to the computational bottleneck (computing Equation 4), so we do not expect this to significantly influence to overall computational cost.

---

> ### Author Response · Authors · 2024-11-27
> **Further Comments on Revision (1/2)**
>
> Thank you again for your work in reviewing our article. We do apologize for the delay in response. We have now completed our revised article, please see the general comment for a summary of the major changes, and our initial individual comment for our preliminary response. Below, we will provide a more detailed response to some of your comments.
>
> > As one of the two cornerstones of the contributions, it would be nice to see where the authors differ in their implementation from previous work that allowed them to compute the PH on the GPU. The details seem to be lacking in the paper and together with the release of the code, would strengthen the paper considerably.
>
> We have added a sentence to emphasize the fact that Equation (4) allows us to parallelize the computation in L204. We also mention this again in L330. As our initial response discussed, the computation of Equation (4) only involves computing min and max of a small distance matrix. We compute Equation (4) for $s$ subsamples in parallel on the GPU.
>
> **Response to comments regarding shape matching experiment:**
>
> We have responded to part of your concern in the initial comments, please let us know if you have further concerns or if you think some of them are not adequately addressed.
>
> Our shape matching experiment is expository in nature, highlighting the behavior of PPM-Reg under the context of GAN training. Compared with shape matching experiments that strive for the best accuracy, the difference in motivation can cause confusion. We have revised the exposition at the beginning of the shape matching experiment to emphasize the goals of the experiment (L367-371).
>
> As our initial response discussed, showing the result when the training loss converges is less relevant. In theory, using a valid metric (such as  EMD) on the space of probability measures with or without regularization will result in the same limit (if $\mu, \nu$ are probability measures such that $d(\mu, \nu) = 0$, then $\mu = \nu$). For simple tasks such as matching unit circles or the union of two intersecting unit circles, any valid metrics would have reasonable performance when converging to zero. Rather, our goal is to show that adding PPM-Reg will emphasize the topological features during the optimization. This is important in GAN settings where the loss function is changing at every iteration due to the change in parameterized projection.
>
> To explore the weakness of only using probability measures to implicitly optimize for topological features, in Appendix E.2, we have added a modified shape matching experiment, which prevents the centroid of the trained shape from converging to the centroid of the reference. This is done to mimic the GAN setting where training algorithms often converge to saddle points rather than global minima. In this modified setting, we empirically show that PPM-Reg is still able to effectively regularize the topological features.
>
> >In the figure the results could be improved by showing how the final output has converged and in two dimensions, which would be natural as other metrics are not provided.
>
> We have added the converged shape in Figure 4.
>
> (Continued on next comment)

---

> ### Author Response · Authors · 2024-11-27
> **Further Comments on Revision (2/2)**
>
> > If the reviewer would have to show to the reader that a certain regularization term is effective, a good strategy would be to take a variety of datasets and a variety of models (perhaps with certain properties) and show that the regularization consistently improves the scores across the board. As the paper is currently presented, the scope of the experiments is rather limited. Moreover, there is also comparable work on topological regularizers for classification problems which would also provide a nice comparison partner. A last potential point of improvement is the connection with other topological methods and regularization terms. The application of persistent homology is not new and a comparison with previous work, both in terms of computational performance and increase in accuracy would also be very nice.
> >
> > Although the reviewer commends the line of work the authors have decided to pursue, a considerable amount of work would have to be done to create a logical and convincing experimental suite. Multiple reference architectures of GAN's would have to be implemented with more standard metrics. Perhaps other tasks would also benefit from this regularization term, such as classification and regression tasks.
>
> Due to the space limitations of the paper, we needed to choose an experimental focus for our article. We choose to consider GANs, where the **explicit** regularization of topological features has not been deeply explored in the topological data analysis community. There have been several recent works which have studied the use of persistent homology as an **evaluation metric** (this motivated is discussed in a new paragraph in the introduction - L37-L53)
> - Jeremy Charlier, Radu State, et al. PHom-GeM: Persistent homology for generative models. Swiss Conference on Data Science (SDS), 2019 IEEE International Conference. IEEE (2019)
> - Sharon Zhou, Eric Zelikman, Fred Lu, Andrew Y. Ng, Gunnar E. Carlsson, and Stefano Ermon. Evaluating the disentanglement of deep generative models through manifold topology. In ICLR (2021)
> - Valentin Khrulkov and Ivan Oseledets. Geometry score: A method for comparing generative adversarial networks, ICML (2018)
> - Serguei Barannikov, Ilya Trofimov, Grigorii Sotnikov, Ekaterina Trimbach, Alexander Korotin, Alexander Filippov, and Evgeny Burnaev. Manifold topology divergence: A framework for comparing data manifolds. NeurIPS (2021)
>
> But there is little to no work on using persistent homology during training; we believe a reason for this fact is that traditional topological methods cannot be scaled to larger tasks such as the ones we have considered (as the reviewer also points out).
>
> We believe there may be many other applications such as classification or regression tasks where our methods may prove beneficial. However, we believe this is beyond the scope of our current article, and would leave this for future interesting work. We have added a sentence in the conclusion (L539) regarding this.
>
> Despite this, we agree with you that it would improve the paper to include further experiments, and we have done our best to include additional experiments within a very limited timeframe and with limited resources.
> - **Additional Experiment - Image Generation (Appendix F.2):** We have added a new image generation experiment for larger (64 x 64) images using the CelebA and LSUN Kitchen datasets. In this larger-scale setting, we observe that our method is still able to provide a significantly performance gain, demonstrating the efficacy of PPM-Reg in larger experiments.
> - **Additional Experiment - Semi-Supervised Learning (Appendix G.2):** We perform an additional SSL experiment with the SVHN dataset with similarly large performance gains.
>
> We agree with your comment in the official review that our methodology can be useful when the topological properties of lower-dimensional submanifolds of a dataset are of importance. One of the motivations of this work is to create efficient and **scalable** tools to enable **explicit** regularization of topological features in larger-scale machine learning tasks.
>
> Let us know if you have further concerns.

---

> > ### Comment · Reviewer_iUNk · 2024-12-02
> >
> > The reviewer would like to thank the authors for their extensive comments and engaging in the discussion. I have raised my score,
> > although fundamentally the hesitance regarding the use of a what seems a non-standard way of evaluating the experimental section remains.

---

> > > ### Author Response · Authors · 2024-12-02
> > > **Response to Reviewer Comment**
> > >
> > > Thank you for taking the time to read through our response. As we have pointed out in our response, we believe we are using standard evaluation metrics in all our experiments:
> > > - **Shape Matching**: Our goal is to show that PPM can appropriately regularize the topology of point clouds, which is measured by the full (not subsampled) persistent homology between the point clouds. We train using distance between PPMs and evaluate using distance between full persistence diagrams.
> > > - **Image Generation**: Our evaluation metrics for image generation are standard methods to evaluate image quality, listed with references on L431-434.
> > > - **Semi-Supervised Learning**: Here, we are just using classification accuracy, which is standard.
> > >
> > > Could you please point out specifically which aspect you feel is non-standard?  Unfortunately, at this point, we will not be able to perform new experiments, but we hope that we can at least clarify our choices. Thank you again for taking part in this review process and suggesting several changes which we believe have improved our article.

---

### Official Review · Reviewer_MDvZ · 2024-11-02

**Soundness:** 3
**Presentation:** 3
**Contribution:** 3
**Rating:** 8
**Confidence:** 3

**Summary:**

This paper proposes to introduce a topological regularization term to the objective function used in the context of GAN.

As standard topological descriptors (known as persistence diagrams) are typically too expensive to compute, they propose to rely on the (recently introduced) notion of "principal persistence measure". Losely speaking, the principal persistence measure is a distribution in the space of persistence diagrams, but its elements have (at most) one point, and can thus be (roughly speaking) identified with distribution on the open half-plane $\{b < d,\ (b,d) \in R^2\}$ augmented with an additionnal virtual point $\star$.

The work then introduce a notion of MMD distance between PPM. The MMD has several benefits: it is fairly easy to compute and induce smooth gradients.

The authors showcase their method in a series of experiments.

**Strengths:**

- Good introduction / preliminaries (section 1 and 2).
- The work mixes results from different ideas (mix of MMD, persistence measures, etc.), and may be useful beyond the scope covered by this paper (improving GAN training). It showcases the use of PPM which are interesting objects in their own, introduce novel possibly useful metric between these topological descriptors, etc.
- interesting experiments, mixing pedagogical Proof-of-Concept and more advanced experiments.
- Nice animation in the supplementary material!

**Weaknesses:**

1. While well written, the introduction and section 2 somewhat fail to motivate the need to account for the geometry/topology when comparing distributions in the latent space. This is just given as a fact (line 142-143), but basically, why should one care about such information? Is there some situations in which this clearly lacks in GAN models? I understand that this needs is empirically confirmed in the experiment sections, but is there a good _a priori_ reason?

2. Somewhat in the same vein: Wasserstein-like distances between topological descriptors are discarded because "gradients are not smooth", but is it a real problem in practice? I would expect stochasting GD to not actually bother which such details.

3. One may argue that GAN tend to be outdated in comparison to more modern generative models (DDPM, etc.). I nonetheless believe that this does not significantly impede the contributions of the work, whose interest is not limited to GAN only.

### Minor remarks (not real weaknesses)

- I believe that the weighting by a factor $\ell$ in the "induced kernel" from $R^2$ to $\Omega$ resemble the PWK kernel of Kusano et al., and more generally the kind of weighting needed in linear representation of persistence diagrams (see Divol&Lacombe, along with Obayashi et al., "Persistence diagrams with linear machine learning models", and related works).

**Questions:**

See question 1. and 2. in the Weaknesses section.

Additionnally :
- I did not have time to read the appendix in details so I may be wrong, but regarding the Remark 1: in theory, PDs may have infinite total mass (here in the work, everything simplify as one consider PDs with at most 1 point), in which case $n,m, N = +\infty$. And even if one restrict to finite diagrams, without putting a uniform bound on their cardinalitty, one may still have $n,m \to \infty$ and (I guess) peculiar behavior of the metric. Does the remark implicitely consider a space of diagrams with uniformly bounded mass/number of points? From a quick glance, $M_{lin}$ only incorporate a standard integrability constraint but nothing about the cardinality of the diagrams.

- The animations show that optimization let few "leftovers", can you comment on this? Is it due to a mix of (gaussian-kernel based) MMD which indeed tends to let leftovers behind + the subsampling in the PPM which needs to be "lucky" to put some gradient on this points? Would using the Energy-distance MMD $k(x,y) = - |x-y|$ improve on this particular aspect? (note: gradient may no longer be smooth using this MMD)

---

> ### Author Response · Authors · 2024-11-25
> **Initial Comment (1/2)**
>
> We would like to thank the reviewer for taking the time to review our article and provide helpful feedback! Firstly, we apologize for the delay in response. We have been taking all of the reviews into consideration and spending considerable time to address all of the comments. Our original plan was to respond after completing a revision of the paper in order to directly point to how our revision addresses each comment. In particular, we have been performing additional experiments as requested by the reviewer, but this had led to some delays due to limited computational resources. In order to have some time for discussions, we provide some initial comments and will provide further detailed comments once our revised article is complete.
>
> > While well written, the introduction and section 2 somewhat fail to motivate the need to account for the geometry/topology when comparing distributions in the latent space. This is just given as a fact (line 142-143), but basically, why should one care about such information? Is there some situations in which this clearly lacks in GAN models? I understand that this needs is empirically confirmed in the experiment sections, but is there a good _a priori_ reason?
>
> In our revised article, we will add a new section in the introduction to clarify the motivation for using topological regularizers in GAN models. In brief, under the manifold hypothesis, latent space matching aims to learn the underlying low-dimensional structure of the dataset. In particular, the union of manifold hypothesis has been empirically verified in
> - Bradley CA Brown, Anthony L Caterini, Brendan Leigh Ross, Jesse C Cresswell, and Gabriel Loaiza-Ganem. Verifying the union of manifolds hypothesis for image data. ICLR (2023)
>
> Furthermore, in the GAN setting, recent work has shown that persistence-based topological methods provide an effective evaluation metric for GANs.
> - Jeremy Charlier, Radu State, et al. PHom-GeM: Persistent homology for generative models. Swiss Conference on Data Science (SDS), 2019 IEEE International Conference. IEEE (2019)
> - Sharon Zhou, Eric Zelikman, Fred Lu, Andrew Y. Ng, Gunnar E. Carlsson, and Stefano Ermon. Evaluating the disentanglement of deep generative models through manifold topology. In ICLR (2021)
> - Valentin Khrulkov and Ivan Oseledets. Geometry score: A method for comparing generative adversarial networks, ICML (2018)
> - Serguei Barannikov, Ilya Trofimov, Grigorii Sotnikov, Ekaterina Trimbach, Alexander Korotin, Alexander Filippov, and Evgeny Burnaev. Manifold topology divergence: A framework for comparing data manifolds. NeurIPS (2021)
>
> These papers suggest that topological information is in fact important in GAN evaluation, and thus motivates explicitly taking topological features into account during training. Our scalable method *enables* the use of such topological methods in GAN training.
>
> > Somewhat in the same vein: Wasserstein-like distances between topological descriptors are discarded because "gradients are not smooth", but is it a real problem in practice? I would expect stochasting GD to not actually bother which such details.
>
> In addition to the theoretical smoothness results, there is a practical reason to use MMD rather than WD as a distance between topological descriptors. In particular, the computational cost of computing MMD is significantly cheaper. We will include an additional experiment which will be added to Table 1 to experimentally compare the computation of WD and MMD.
>
> > One may argue that GAN tend to be outdated in comparison to more modern generative models (DDPM, etc.). I nonetheless believe that this does not significantly impede the contributions of the work, whose interest is not limited to GAN only.
>
> We agree that GANs may be slightly outdated in terms of generative models. However, we chose to use GANs as our main example to demonstrate the fact that our methods can be applied to larger scale experiments (where the cost of traditional persistent homology may already be prohibitive). It would be interesting to consider the application of these methods to more modern generative models, and leave this as future work.
>
> (Continued in next comment)

---

> > ### Author Response · Authors · 2024-11-25
> > **Initial Comments (2/2)**
> >
> > > I believe that the weighting by a factor $\ell$ in the "induced kernel" from $\mathbb{R}^2$ to $\Omega$ resemble the PWK kernel of Kusano et al., and more generally the kind of weighting needed in linear representation of persistence diagrams (see Divol&Lacombe, along with Obayashi et al., "Persistence diagrams with linear machine learning models", and related works).
> >
> > We thank the reviewer for pointing out these references! The authors were unaware of the results in these articles, and as the reviewer correctly pointed out, our kernel is in fact related to the PWK kernel of Kusano (2016). We believe this is still a novel insight in terms of application in terms of the PPM, but acknowledge that these results can be adapted from the theory in these papers (though our Theorem 1 is slightly more general than Kusano -- see next comment). We will make changes to proper citations to these papers throughout. To the authors' knowledge, the statement and proof of Theorem 3 is novel.
> >
> > > I did not have time to read the appendix in details so I may be wrong, but regarding the Remark 1: in theory, PDs may have infinite total mass (here in the work, everything simplify as one consider PDs with at most 1 point), in which case $n, m, N = + \infty$ . And even if one restrict to finite diagrams, without putting a uniform bound on their cardinalitty, one may still have $n, m \to \infty$ and (I guess) peculiar behavior of the metric. Does the remark implicitely consider a space of diagrams with uniformly bounded mass/number of points? From a quick glance, $M_{lin}$ only incorporate a standard integrability constraint but nothing about the cardinality of the diagrams.
> >
> > Kusano considers the case of finite persistence diagrams, while our characteristicness result (Corollary 1, Theorem 5) allows for the case of possibly infinite cardinality (assuming the integrability constraint). The integrability constraint is a standard requirement for infinite cardinality diagrams (also discussed in Divol Lacombe for instance) such that the total *persistence* is bounded.
> >
> > > The animations show that optimization let few "leftovers", can you comment on this? Is it due to a mix of (gaussian-kernel based) MMD which indeed tends to let leftovers behind + the subsampling in the PPM which needs to be "lucky" to put some gradient on this points? Would using the Energy-distance MMD $k(x,y) = -|x-y|$ improve on this particular aspect? (note: gradient may no longer be smooth using this MMD)
> >
> > Our main goal with the shape matching section was to demonstrate the effectiveness of PPM-Reg *during* the optimization, and not necessarily show convergence. Thus, we only showed plots and animations for the initial part of the optimization. However, we will add plots and animations to show convergence. The reviewer is correct in that the "leftovers" are influenced by the choice in the underlying loss function, and our choice of underlying loss functions is motivated by the loss functions used in the GAN experiments.

---

> > > ### Comment · Reviewer_MDvZ · 2024-11-25
> > > **Thanks**
> > >
> > > Thank you for your comments.

---

> ### Author Response · Authors · 2024-11-27
> **Further Comments on Revision**
>
> Thank you again for your work in reviewing our article. We do apologize for the delay in response. We have now completed our revised article, please see the general comment for a summary of the major changes, and our initial individual comment for our preliminary response. Below, we will provide a more detailed response to some of your comments.
>
> > While well written, the introduction and section 2 somewhat fail to motivate the need to account for the geometry/topology when comparing distributions in the latent space. This is just given as a fact (line 142-143), but basically, why should one care about such information? Is there some situations in which this clearly lacks in GAN models? I understand that this needs is empirically confirmed in the experiment sections, but is there a good _a priori_ reason?
>
> We have added a new paragraph **Topological Features of Latent Representations** (L37-53) at the beginning of the introduction to better motivate the use of topological methods in GANs. In summary, under the manifold hypothesis, we assume that datasets are often concentrated about lower-dimensional manifolds, motivating the fact that GANs should aim to capture the geometric and topological properties of these lower-dimensional representations. In fact, recent work has shown that persistence-based topological metrics provide a highly effective evaluation metric for GANs. While metrics between probability measures **implicitly** capture this information, these articles motivated us to try **explicitly** regularizing for the topological structure.
>
> > Somewhat in the same vein: Wasserstein-like distances between topological descriptors are discarded because "gradients are not smooth", but is it a real problem in practice? I would expect stochasting GD to not actually bother which such details.
>
> We have added an experimental comparison between WD and MMD to Table 1 (L401).
>
> > I believe that the weighting by a factor $\ell$ in the "induced kernel" from $\mathbb{R}^2$ to $\Omega$ resemble the PWK kernel of Kusano et al., and more generally the kind of weighting needed in linear representation of persistence diagrams (see Divol&Lacombe, along with Obayashi et al., "Persistence diagrams with linear machine learning models", and related works).
>
> Thank you again for pointing this out. We have now added references to these papers throughout. In particular, we have cited them when discussing the novelty of our theoretical results (L92-93), when introducing our kernel (L230), the fact that one can adapt their methods to prove a result similar to our Theorem 1 (L255), and their related stability result (L274). However, we have opted to keep our proof of Theorem 1 in the appendix. We believe that our proof provides some additional subtle details, which were not immediately obvious to us (whereas Kusano (2016) simply states the result as being true).
>
> >I did not have time to read the appendix in details so I may be wrong, but regarding the Remark 1: in theory, PDs may have infinite total mass (here in the work, everything simplify as one consider PDs with at most 1 point), in which case $n, m, N = + \infty$ . And even if one restrict to finite diagrams, without putting a uniform bound on their cardinalitty, one may still have $n, m \to \infty$ and (I guess) peculiar behavior of the metric. Does the remark implicitely consider a space of diagrams with uniformly bounded mass/number of points? From a quick glance, $M_{lin}$ only incorporate a standard integrability constraint but nothing about the cardinality of the diagrams.
>
> We have now edited Remark 1 (L278-282) to clarify this point.
>
> Let us know if you have further concerns.

---

### Official Review · Reviewer_BKqM · 2024-11-06

**Soundness:** 3
**Presentation:** 2
**Contribution:** 3
**Rating:** 8
**Confidence:** 3

**Summary:**

The paper proposes a regularizer to be used in Generative Adversarial Networks (GANs) in addition to the “standard” distance between distributions (e.g. Wasserstein distance). The regularizer consists in computing the probability of persistent homology diagrams through Principal Persistent Measures (PPM) on both real and generated data and comparing them through MMD. The paper provides a few theorems proving the smoothness of the MMD applied to compare PPM under mild conditions and experimentally shows the benefits of this regularizer on both generative metrics and semi-supervised classification tasks.

**Strengths:**

The paper pairs the proposed regularizer with a solid theoretical analysis of the smoothness of the resulting loss function, which justifies its use as an additional term to the (W)GAN loss.

Experimental results show that the proposed regularizer provides a significant advantage in training GANs, especially for self-supervised tasks.

Although the paper is theoretically dense, it can still be followed by non-experts in the PH fields. Nevertheless, it would be helpful to remark the purpose of each theorem, or sequence of theorems, before introducing them.

**Weaknesses:**

The main weakness of the paper is the introduction and motivation of the problem and the relation of the proposed method to the current literature. From the introduction (and also the title), it isn’t really clear what problem the paper is going to tackle. While the introduction talks about general regularizers for latent space representations, most of the methodological development and the experimental section tackle specifically the problem of regularizing GANs. I would be clearer about this from the beginning of the paper or show applications beyond GANs (beyond the shape-matching toy example).

I would also better frame the work in the context of GANs, and regularization strategies for GANs. A few methods are mentioned in the literature, but none is discussed in depth. For instance, PHom-GeM seems to share objectives and strategies similar to those of the proposed work. It would be ideal to compare also with some of these methods.

Also, it is not clear what theorems are novel, and which ones are just reported or adapted from results in the existing literature. Maybe they are all novel, but specifying the current state of the literature and which gaps need to be filled (from a theoretical point of view) would help to better understand the contribution.

**Questions:**

I would like the authors to clarify the points I have highlighted on the weaknesses, especially the contribution and the relation with other GAN regularizers.

Moreover, the method proposes the use of MMD rather than WD to compare the PPMs. Is there any theoretical and/or practical reason? Would still be possible to use WD and experimentally compare the performance of WD and MMD?

Minor comments:
- The citation format is weird and interferes with the reading.
- At row 66, what would be the classic PH pipeline?
- In row 155, you provide 5 introductory references to PH, these are far too many, and this does not really help the reader.
- Row 383: “consider provide”
- Table 1: since you computed averages over 10 runs, it would make sense to report also the standard deviation.

---

> ### Author Response · Authors · 2024-11-25
> **Initial Comment**
>
> We would like to thank the reviewer for taking the time to review our article and provide helpful feedback! Firstly, we apologize for the delay in response. We have been taking all of the reviews into consideration and spending considerable time to address all of the comments. Our original plan was to respond after completing a revision of the paper in order to directly point to how our revision addresses each comment. In particular, we have been performing additional experiments as requested by the reviewer, but this had led to some delays due to limited computational resources. In order to have some time for discussions, we provide some initial comments and will provide further detailed comments once our revised article is complete.
>
> > The main weakness of the paper is the introduction and motivation of the problem and the relation of the proposed method to the current literature. From the introduction (and also the title), it isn’t really clear what problem the paper is going to tackle. While the introduction talks about general regularizers for latent space representations, most of the methodological development and the experimental section tackle specifically the problem of regularizing GANs. I would be clearer about this from the beginning of the paper or show applications beyond GANs (beyond the shape-matching toy example).
>
> Our article proposes a general methodology for scalable topological regularization, but our article focuses on its applications in the GAN setting, which is a setting in which our scalable methodology *enables* the use of topological regularizers. In our revised article, we will rewrite the initial part of our introduction to clarify this point and to better motivate the use of such topological regularization in GANs.
>
> > I would also better frame the work in the context of GANs, and regularization strategies for GANs. A few methods are mentioned in the literature, but none is discussed in depth. For instance, PHom-GeM seems to share objectives and strategies similar to those of the proposed work. It would be ideal to compare also with some of these methods.
>
> As mentioned, we will include a new section in the introduction to motivate topological regularization in GANs. You are correct that PHom-GeM has similar objectives, but they only use topological methods as an *evaluation metric* for GANs. In fact, this article, along with a few others have suggested using persistent homology as an evaluation metric:
> - Sharon Zhou, Eric Zelikman, Fred Lu, Andrew Y. Ng, Gunnar E. Carlsson, and Stefano Ermon. Evaluating the disentanglement of deep generative models through manifold topology. In ICLR (2021)
> - Valentin Khrulkov and Ivan Oseledets. Geometry score: A method for comparing generative adversarial networks, ICML (2018)
> - Serguei Barannikov, Ilya Trofimov, Grigorii Sotnikov, Ekaterina Trimbach, Alexander Korotin, Alexander Filippov, and Evgeny Burnaev. Manifold topology divergence: A framework for comparing data manifolds. NeurIPS (2021)
>
> This forms one of the primary motivations for using a topological regularizer in the loss function for training GANs.
>
> > Also, it is not clear what theorems are novel, and which ones are just reported or adapted from results in the existing literature. Maybe they are all novel, but specifying the current state of the literature and which gaps need to be filled (from a theoretical point of view) would help to better understand the contribution.
>
> We will clarify the novelty of the theorems in the revised version. In particular, another reviewer (MDvZ) has pointed out results in two references we were not aware of:
> - Genki Kusano, Yasuaki Hiraoka, Kenji Fukumizu, Persistence weighted Gaussian kernel for topological data analysis , ICML (2016)
> - Vincent Divol, Theo Lacombe, Understanding the topology and the geometry of the space of persistence diagrams via optimal partial transport, Journal of Applied and Computational Topology (2021)
>
> While the statements of Theorems 1-2 are novel in the context of PPMs (and our Theorem 1 is slightly more general), we will make changes to proper citations to these papers throughout. To the knowledge of the authors, Theorem 3 is novel.
>
> > Moreover, the method proposes the use of MMD rather than WD to compare the PPMs. Is there any theoretical and/or practical reason? Would still be possible to use WD and experimentally compare the performance of WD and MMD?
>
> The theoretical reason is that MMD has continuous gradients (Theorem 3), and the practical reason is the computational cost. We will include an additional experiment which will be added to Table 1 to experimentally compare the computation of WD and MMD.
>
> We will also address your minor comments, and directly refer to our revised article once it is complete.

---

> ### Author Response · Authors · 2024-11-27
> **Further Comments on Revision**
>
> Thank you again for your work in reviewing our article. We do apologize for the delay in response. We have now completed our revised article, please see the general comment for a summary of the major changes, and our initial individual comment for our preliminary response. Below, we will provide a more detailed response to some of your comments.
>
> > The main weakness of the paper is the introduction and motivation of the problem and the relation of the proposed method to the current literature. From the introduction (and also the title), it isn’t really clear what problem the paper is going to tackle. While the introduction talks about general regularizers for latent space representations, most of the methodological development and the experimental section tackle specifically the problem of regularizing GANs. I would be clearer about this from the beginning of the paper or show applications beyond GANs (beyond the shape-matching toy example).
> > I would also better frame the work in the context of GANs, and regularization strategies for GANs. A few methods are mentioned in the literature, but none is discussed in depth. For instance, PHom-GeM seems to share objectives and strategies similar to those of the proposed work. It would be ideal to compare also with some of these methods.
>
> We now immediately focus on GANs and have added a new paragraph **Topological Features of Latent Representations** (L37-53) at the beginning of the introduction to better motivate the use of topological methods in GANs. In summary, under the manifold hypothesis, we assume that datasets are often concentrated about lower-dimensional manifolds, motivating the fact that GANs should aim to capture the geometric and topological properties of these lower-dimensional representations. In fact, recent work has shown that persistence-based topological metrics provide a highly effective evaluation metric (including the PHom-GeM paper you mentioned) for GANs. While metrics between probability measures **implicitly** capture this information, these articles motivated us to try **explicitly** regularizing for the topological structure.
>
> Due to the space limitations of the paper, we wanted to focus our introductory motivation section on such topological methods in GANs. In particular, our article is focused on developing the tools to perform such explicit topological regularization, and to empirically test whether this can improve performance. While there may be other useful GAN regularization methods available, we believe this is beyond the scope of our article.
>
>  >Also, it is not clear what theorems are novel, and which ones are just reported or adapted from results in the existing literature. Maybe they are all novel, but specifying the current state of the literature and which gaps need to be filled (from a theoretical point of view) would help to better understand the contribution.
>
>  We have now added a sentence (L92-93) to address this.
>
> > Moreover, the method proposes the use of MMD rather than WD to compare the PPMs. Is there any theoretical and/or practical reason? Would still be possible to use WD and experimentally compare the performance of WD and MMD?
>
> We have added an experimental comparison between WD and MMD to Table 1 (L401)
>
> > The citation format is weird and interferes with the reading.
>
> We have now changed the citation format.
>
> > At row 66, what would be the classic PH pipeline?
>
> We have removed this sentence, but have still explained the standard pipeline in L58-60.
>
> > In row 155, you provide 5 introductory references to PH, these are far too many, and this does not really help the reader.
>
> We have now reduced this to two references.
>
> > Row 383: “consider provide”
>
> Thank you for catching this! Now fixed.
>
> > Table 1: since you computed averages over 10 runs, it would make sense to report also the standard deviation.
>
> We have now added the standard deviation to Table 1.
>
> Let us know if you have further concerns.

---

> > ### Comment · Reviewer_BKqM · 2024-12-02
> >
> > I thank the authors for answering my main concerns. Now, the scope of the paper and its contribution seems more clear to me. I have raised my score accordingly.

---

> > > ### Author Response · Authors · 2024-12-02
> > > **Reponse to Reviewer Comment**
> > >
> > > Thank you for taking the time to read through our responses. We are glad to hear that we were able to address your main concerns, and appreciate the increased score. Your comments have been incredibly helpful in improving the quality of our article.

---

### Author Response · Authors · 2024-11-27
**Overview of Changes in Revision**

We would like to thank all of the reviewers for taking the time to carefully review our article, and to provide helpful feedback which has certainly improved the quality of our article. We do apologize for the delay in response. Here, we provide an overview to the major changes made in our revision, and will respond to each reviewer individually to discuss how these changes address their concerns.

### Section 1 and Section 2
- (L37-L53) A new paragraph **Topological Features of Latent Representations** is added at the beginning of the introduction to better motivate the need for capturing topological features in GAN settings. Previously, portions of this paragraph were scattered throughout Section 1 and 2. As suggested by the reviewers, we believe having a coherent motivation at the outset clarifies the need for our proposed method.
- (L92-L93) We included a sentence clarifying the novelty of our theoretical results.
- Section 2: We have shortened this section as some of the discussion has been moved to the introduction.

### Section 3 - Section 5
There are only minimal changes to these sections. As pointed out by Reviewer MDvZ, the article
- Genki Kusano, Yasuaki Hiraoka, Kenji Fukumizu, Persistence weighted Gaussian kernel for topological data analysis , ICML (2016)

has previously suggested the use of related kernel methods for persistence diagrams, and we have added the appropriate citations throughout.

### Experiments
- (L367-371) We have added some exposition to clarify the main goals of the shape matching experiment.
- (L401, Table 1) As suggested by reviewers, we have empirically compared the performance of PPM-Reg using the Wasserstein distance rather than MMD. In particular, as the number of subsamples (and thus numbers of points in the PPM) grow, the Wasserstein distance quickly becomes expensive to compute.
- **Additional Experiment - Shape Matching (Appendix E.2):** We have added a modified shape matching experiment, which prevents the centroid of the trained shape from converging to the centroid of the reference. This is done to mimic the GAN setting where training algorithms often converge to saddle points rather than global minima. In this modified setting, we empirically show that PPM-Reg is still able to effectively regularize the topological features.
- **Additional Experiment - Image Generation (Appendix F.2):** We have added a new image generation experiment for larger (64 x 64) images using the CelebA and LSUN Kitchen datasets. In this larger-scale setting, we observe that our method is still able to provide a significantly performance gain, demonstrating the efficacy of PPM-Reg in larger experiments.
- **Additional Experiment - Semi-Supervised Learning (Appendix G.2):** We perform an additional SSL experiment with the SVHN dataset with similarly large performance gains.


**Edit (Nov. 27, 2024):** We have made minor changes to the revised article -- mainly fixing typos and reorganizing some material in Appendices E, F, G. All line references above are still correct.

---

### Meta-Review · Area_Chair_HnHH · 2024-12-21

**Metareview:**

This paper suggests an efficient and scalable alternative to persistent homology (PH) for topological regularizers. In particular it suggests parallelizing PH with mini-batches (Principal Persistence Measures) and replacing the Wasserstein distance with Maximum Mean Discrepancy. The paper's main application focus is on generative adversarial networks (GANs) where the topological regularizer is added to the loss function.

The reviewers overall appreciated the contribution of this paper that provides a justified yet scalable regularizer to compare distribution in latent spaces, which could potentially have applications beyond the GAN setting tested in this paper. Some concerns of the reviewers included lacking motivation for the need to account for topology when comparing distributions, insufficient accessible introduction and motivation (for non-experts), unclear which theorems are novel, and some concerns regarding the limited experimental setup. The authors have addressed the majority of these concerns to a satisfactory level during rebuttal.

**Additional Comments On Reviewer Discussion:**

No additional comments.

---

### Decision · Program_Chairs · 2025-01-22

Accept (Poster)